# Interpretable Mixture of Experts

Aya Abdelsalam Ismail[*1], Sercan Ö. Arik[2], Jinsung Yoon[2], Ankur Taly[2], Soheil Feizi[3], and Tomas Pfister[2]

[1]Genentech
[2]Google Cloud AI
[3]University of Maryland

**Reviewed on OpenReview:** <https://openreview.net/forum?id=DdZoPUPm0a>

## Abstract

The need for reliable model explanations is prominent for many machine learning applications, particularly for tabular and time-series data as their use cases often involve high-stakes decision making. Towards this goal, we introduce a novel interpretable modeling framework, Interpretable Mixture of Experts (IME), that yields high accuracy, comparable to 'black-box' Deep Neural Networks (DNNs) in many cases, along with useful interpretability capabilities. IME consists of an assignment module and a mixture of experts, with each sample being assigned to a single expert for prediction. We introduce multiple options for IME based on the assignment and experts being interpretable. When the experts are chosen to be interpretable such as linear models, IME yields an inherently-interpretable architecture where the explanations produced by IME are the exact descriptions of how the prediction is computed. In addition to constituting a standalone inherently-interpretable architecture, IME has the premise of being integrated with existing DNNs to offer interpretability to a subset of samples while maintaining the accuracy of the DNNs. Through extensive experiments on 15 tabular and time-series datasets, IME is demonstrated to be more accurate than single interpretable models and perform comparably with existing state-of-the-art DNNs in accuracy. On most datasets, IME even outperforms DNNs, while providing faithful explanations. Lastly, IME's explanations are compared to commonly-used post-hoc explanations methods through a user study – participants are able to better predict the model behavior when given IME explanations, while finding IME's explanations more faithful and trustworthy.

## 1 Introduction

Tabular and time-series data appear in numerous applications, including healthcare, finance, retail, environmental sciences and cybersecurity, and constitute a major portion of the addressable artificial intelligence market (Chui et al., 2018). Although simple interpretable models like linear regression (LR) or decision trees (DTs) or ARIMA had dominated the real-world applications with these data types, deep neural networks (DNNs) have recently shown state-of-the-art performance, often significantly improving the interpretable models (Arik & Pfister, 2021; Lim et al., 2021; Popov et al., 2019; Joseph, 2021; Gorishniy et al., 2021; Wen et al., 2018), and now they are being used more commonly. Yet, one challenge hindering the widespread deployment of DNNs is their black-box nature (Lipton, 2017) — humans are unable to understand their complex decision-making process. For most tabular and time-series data applications, explainability is crucial (Caruana et al., 2015; Lipton, 2018) — physicians need to understand why a drug would help, and retail data analysts need to gain insights on the trends in the predicted sales, and bankers need to understand why a certain transaction is categorized as fraudulent.

In an attempt to produce explanations, various interpretable architectures have been proposed. Attention-based DNNs (Arik & Pfister, 2021; Lim et al., 2021) constitute one approach with attention weights being

---

*Corresponds to: ismail.aya@gene.com; work done while at Google.

used as explanations; however, recent works (Bibal et al., 2022) show the limitations of such explanations. Differentiable neural versions of generalized additive models (Agarwal et al., 2020; Chang et al., 2021) and soft DTs (Luo et al., 2021) are effective for tabular data but they do not generalize to time-series data. Trend-seasonality decomposition-based architectures (Oreshkin et al., 2019) are proposed for univariate time-series data but can not be used for tabular or multivariate time-series data. Recently, Puri et al. propose DNNs with explanations in the form of continued fractions, whose interpretation complexity increases with the addition of network layers. The above methods have one or more of the following problems: (a) yielding unreliable explanations; (b) yielding complex explanations; (c) can not generalize to different structured data (i.e, both time series and tabular); (d) have varying amounts of accuracy degradation as they trade model interpretability for accuracy. Overall, a systematic interpretable framework while preserving accuracy, remains to be an important direction.

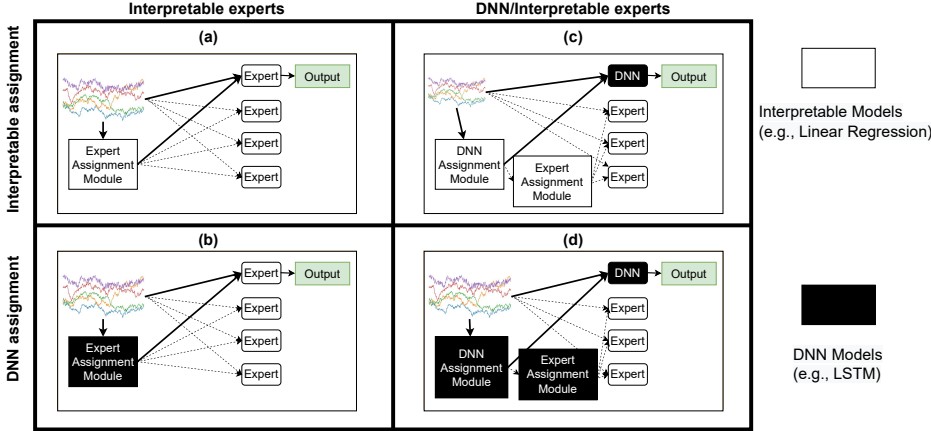

**Figure 1:** Interpretable Mixture of Experts (IME) framework for time-series and tabular data, with single-level and hierarchical assignment options. Solid lines denote the selected assignments. (a) shows **S-IME**$_i$, a single-level assignment IME where both the assignment module and the experts are interpretable. (b) shows **S-IME**$_d$, a single-level assignment IME where the assignment module is a DNN, and interpretability is provided only via the experts. For hierarchical assignment, the primary assignment module chooses between a pre-trained black box expert or S-IME$_{i/d}$, offering partial interpretability by giving insights into which samples benefit from the complexity of black box models. (c) shows **H-IME**$_i$, a hierarchical IME where both the assignment and expert modules are interpretable. (d) shows **H-IME**$_d$, a hierarchical IME where assignment modules are DNNs and experts are interpretable.

We propose a novel framework for inherently-interpretable modeling for tabular and time-series data, with the idea of combining multiple models, that can all or partially be interpretable, in a mixture of experts (ME) framework. ME frameworks are composed of multiple "experts" and an assignment module that decides which expert to pick for each sample. Recent works (Shazeer et al., 2017; Fedus et al., 2021) used ME to replace DNN layers for efficient capacity scaling. On the other hand, we propose employing a single ME to fit different subsets by interpretable experts. Although complex distributions cannot be fit by simple interpretable models, we postulate that small subsets of them can be. Building upon this idea, we show that IME can preserve accuracy while providing useful interpretability capabilities by replacing black-box models with multiple interpretable models. Enabled by innovations in its design, IME is a novel framework with interpretable models that can partially/fully replace or encapsulate DNNs to achieve on-par with or better accuracy, based on the desired explainability-accuracy trade-off. Explanations produced by IME are faithful to the model – we show through user studies that IME explanations are easier to interpret and users tend to trust the explanations produced by IME over those produced by saliency methods.

To address interpretability needs of different applications,we propose multiple options for IME: Single-level assignment S-IME$_i$ & S-IME$_d$ (Fig.1 (a) & (b)) employs an assignment module, either interpretable (yielding both assignment and expert interpretability) or a DNN (yielding only expert interpretability) with interpretable experts. Hierarchical assignment H-IME$_i$ & H-IME$_d$ (Fig.1 (c) & (d)) first selects between a *pre-trained* black-box expert and an IME, and then between different interpretable experts. S-IME$_i$ (Fig.1

(a)) generates explanations in the form of the exact description of predictions as a concise formula, enabling its use in high-stakes applications that require precise explanations. S-IME$_d$ (Fig.1 (b)) can be used for local interpretability, a capability to gain insights on the group of predictions in each cluster. H-IME$_i$ & H-IME$_d$ (Fig.1 (c) & (d)) offer interpretable decision making for a subset of samples while maintaining accuracy. As an example, for a real-world Retail task, approximately 40% of samples can be assigned to an interpretable expert while preserving the same accuracy. The trade-off can be adjusted by the user based on the application needs.

## 2 IME Framework

### 2.1 Overall design

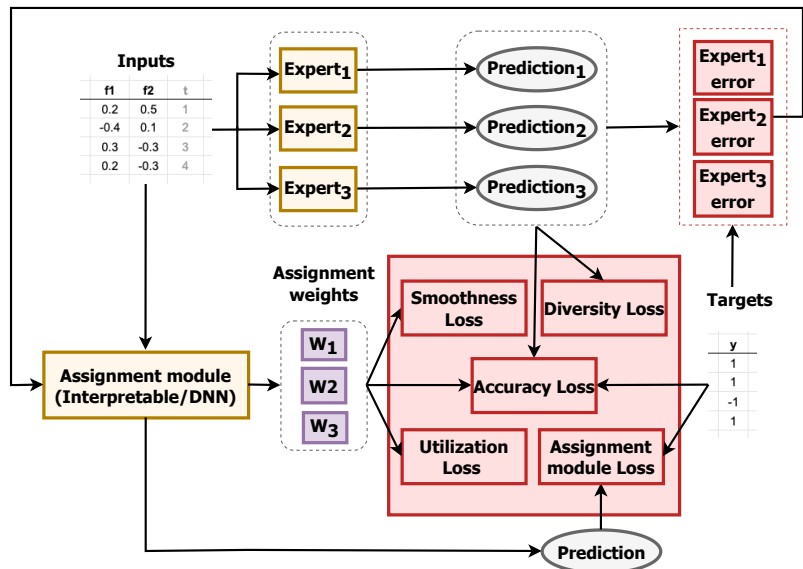

**Figure 2:** S-IME consists 'interpretable experts' (3 in this example) and an 'assignment module'. The input to IME can be time-series or tabular data. Each expert produces a single prediction. The input to the assignment module is both a feature embedding and the past error made by the experts (when the time dimension exists). The assignment module selects a single expert during inference to make the final prediction. We define various losses (shown in red boxes) that are used to supervise the assignment module and expert weights.

IME consists of a set of 'interpretable experts' and an 'assignment module', as overviewed in Fig. 2 (Note that this shows S-IME, a detailed overview of H-IME is shown in the Appendix A Fig. 9). The experts can be any interpretable differentiable model, each with its own trainable weights. They can be different architectures (assuming they accept inputs and generate outputs of particular sizes), e.g. one expert can be linear regression (LR) while the other can be a soft decision tree (DT) (Frosst & Hinton, 2017; Luo et al., 2021; Irsoy et al., 2012). The only requirement for them is being differentiable (for which (Ghods & Cook, 2021) provides an overview). The assignment module can be either an interpretable model or a DNN, as shown in Fig. 1. When we have a pre-trained DNN as one of the experts (in H-IME$_i$ and H-IME$_d$), we employ a hierarchical assignment structure so that the first assignment module (*DNN assignment module*) decides whether a sample should be assigned to a DNN or IME. If the assignment is not a DNN, a second (*expert assignment module*) selects the interpretable model to assign. Furthermore, we introduce a mechanism to control the use of the pre-trained DNN, enabling an accuracy-interpretability trade-off (see Sec. 3.3). Hierarchical assignment is essential with a pre-trained DNN, since adding a pre-trained DNN as an expert candidate (along with other interpretable experts) biases the assignment module, which may lead to choosing the DNN over other experts. During training, the assignment module outputs the selection likelihood for each expert and the target prediction. During inference, the most probable expert is selected. IME can be used for tabular data (where a sample consists of a single observation at a given timestep) or time-series (where a sample consists of multiple observations over a time period).

**Notation.** Broadly, consider a regression problem for input data with $S$ samples $\{(X_i, Y_i)\}_{i=1}^{S}$; with $X = [x_1, \ldots, x_T] \in \mathbb{R}^{N \times T}$, where $T$ is the number of timesteps (for tabular data, we have $T = 1$) and $N$ is the number of features. Outputs are $Y = [y_1, \ldots, y_H] \in \mathbb{R}^H$, where $H$ is the horizon (for tabular data $H = 1$). The inputs until timestep $t$ are denoted as $X_t$. For an IME with $n$ experts, each expert is denoted as $f_i$ with $i$ being the expert index. The predictions made by the expert $i$ are given as $f_i(X) = \hat{Y}_i$ and the corresponding prediction error, for which we use mean squared error (MSE), at a given horizon $h$, is denoted as $e_{i,h} = \frac{1}{S} \sum_{i=1}^{S} (y_h - \hat{y}_h)^2$. The errors made by all experts at time $t$ are denoted as $E_t = [e_{1,t}, \cdots, e_{n,t}]$. The assignment module has two functions: (a) $A_y$ outputs prediction $\hat{Y}_A$. (b) $A_w$ outputs $n$-dimensional vector representing the weight $w_i$ as the probability of choosing a particular expert such that $\sum_{i=1}^{n} w_i = 1$. Both $A_y$ and $A_w$ share the same weights except for the last layer. Note that this can be generalized to classification by simply changing the model outputs.

## 2.2 Learning from past errors

IME can be applicable to general tabular data without any time-dependent feature. However, with time-dependent features available, IME benefits from incorporating past errors (with the accuracy improvements shown in Sec. 4.5). For most datasets, the input-output relationships between consecutive timesteps are relevant. This is often due to consecutive timesteps having overlapping information, or external conditions not changing rapidly. For data with time information, we propose to use past errors as informative signals for the assignment. Specifically, concatenation of both the feature embedding $X_t$ and past error made by the experts $E_{t-1}$, $\overline{X_t} = [X_t; E_{t-1}]$, is used as the input to the assignment module so that it can take into account of the errors while assigning weights for the next timestep. Note that with causal masking, we ensure that the forecasting error never overlaps with the forecasting horizon. For example, if we are forecasting horizon $H$ from timesteps $t + 1$ to $H + t + 1$, only the expert errors until timestep $t$ are added to the input of the assignment module.

## 2.3 Training objectives

IME training is based on the following desiderata:

- Overall model accuracy should be maximized.
- The assignment module should be accurate in selecting the most appropriate experts.
- Utilization of individual experts shouldn't be imbalanced, avoiding all samples being assigned to one expert.
- Experts should yield diverse predictions, helping the assignment module to better discriminate between them.
- For inputs with time information, assignments should be smooth for consecutive inputs, for better generalization and interpretability.

Correspondingly, we propose the following objective functions:

$$
\mathcal{L}\left(f, A_w, A_y, X, \overline{X}, Y\right) = \mathcal{L}_{pred}\left(f, A_w, X, \overline{X}, Y\right) + \beta \mathcal{L}_{util}\left(A_w, \overline{X}\right) + \gamma \mathcal{L}_{div}\left(f, X\right) + \\
\delta \mathcal{L}_{smooth}\left(A_w, \overline{X}\right) + \lambda \mathcal{L}_{A_y}\left(A_y, \overline{X}, Y\right), \tag{1}
$$

where $\beta, \gamma, \delta$ and $\lambda$ are hyperparameters. $\mathcal{L}_{pred}, \mathcal{L}_{util}, \mathcal{L}_{div}, \mathcal{L}_{smooth}$ and $\mathcal{L}_{A_y}$ are prediction accuracy, expert utilization, expert diversity, assignment smoothness and assignment accuracy losses, respectively, explained in detail below.

**Prediction accuracy.** As in (Jacobs et al., 1991), we propose the log likelihood loss under a Mixture of Gaussians assumption:

$$
\mathcal{L}_{pred}(f, A_w, X, \overline{X}, Y) = -\log \sum_{i=1}^{n} A_w(\overline{X})_i \frac{e^{-\|Y - f_i(X)\|^2/2}}{\sqrt{2\pi}}. \tag{2}
$$

This loss can help encourage expert specialization by comparing each expert separately with the target to reduce the average of all these discrepancies. Here, we consider a regression problem. For classification, the loss in Eq. 2 can be adjusted accordingly, Appendix B includes more details.

**Expert utilization.** The assignment module can converge to a state where it produces large assignment weights for the same few experts, resulting in some experts being trained more rapidly and thus selected more. To circumvent this, (Eigen et al., 2013) uses hard constraints at the beginning of training to avoid local minimum, (Bengio et al., 2015) uses soft constraints on the batch-wise average of each expert, and (Shazeer et al., 2017) encourages all experts to have equal importance (i.e, uniform expert utilization) by penalizing the coefficient of variation between different expert utilization. For IME, we propose that each expert should focus on a subset of the distribution. These subsets do not have to be equal in size, so all experts should be utilized but not necessarily in a uniform manner. Given this, we propose the following utilization objective:

$$\mathcal{L}_{util} = (1/N) \sum_{i=1}^{N} e^{-kU_i} - e^{-k}, \tag{3}$$

where $U_i$ is the utilization of expert $i$ such that $U_i = 1/N \sum_{j=1}^{N} w_{i,j}$ and $k$ is a hyperparameter to encourage utilization without enforcing uniformity across experts.

**Expert diversity.** Experts should specialize in different data subspaces (i.e., they would have different expertise), as specialization helps the assignment module in choosing the correct expert for a given data point resulting in better accuracy. Diversity between the outputs of different experts encourages expert specialization. We encourage diversity by employing a contrastive loss:

$$\mathcal{L}_{div}(f, X) = - \sum_{i=1}^{n} \log \frac{\exp(S(f_i(X), f_i(X+\eta))/\tau)}{\sum_{k=1}^{n} \mathbb{1}_{[k \neq i]} \exp(S(f_i(X), f_k(X))/\tau)}, \tag{4}$$

where $S(u, v) = u^T v / \|u\| \|v\|$ denote the dot product between $l_2$ normalized $u$ and $v$, $\mathbb{1}_{[k \neq i]} \in \{0, 1\}$ is the indicator function such that 1 iff $k \neq i$, and $\tau$ is the temperature parameter. We propose it as minimizing the distance between outputs from the same expert and maximizing the distance between outputs from different experts. We add a Gaussian noise $\eta$ with zero mean and unit variance to the inputs, and define positive pairs as outputs coming from the same expert (with and without the noise) and negative pairs as outputs coming from two different experts (both without noise).

**Assignment smoothness.** For data with time component, consecutive inputs should have mostly overlapping information, so one would expect mostly similar assignment for them. To promote smooth transition of assignments over time, we adopt the Kullback–Leibler (KL) divergence (Kullback & Leibler, 1951) between the weights output by the assignment module for consecutive timesteps $t-1$ and $t$:

$$\mathcal{L}_{smooth}\left(A_w, \overline{X}\right) = D_{\mathrm{KL}}\left(A_w\left(\overline{X}_{t-1}\right) \| A_w(\overline{X}_t)\right), \tag{5}$$

where $D_{\mathrm{KL}}(P \| Q)$ denote the KL divergence between two distributions $P$ and $Q$ defined on the same probability space. Here, we assume a forecasting horizon of 1, and this can be modified for longer forecasting horizons by summing over forecasting timesteps. Assignment smoothness can also be helpful for improving the interpretability, as users can build more reliable insights.

**Assignment module accuracy.** The assignment module is designed to generate predictions (that are only used during training) along with the weights, and we propose to minimize the following error for superior assignment accuracy:

$$\mathcal{L}_{A_y}\left(A_y, \overline{X}, Y\right) = (1/N) \sum_{i=1}^{N} \left(Y - A_y\left(\overline{X}\right)\right)^2. \tag{6}$$

**H-IME regularization.** In H-IME to avoid all samples being assigned to the pre-trained DNN expert, an additional regularization term is added to Eq. 1: $\alpha U_{\mathrm{DNN}}$, where $U_{\mathrm{DNN}}$ is the DNN expert utilization.

$$U_{\mathrm{DNN}} = (1/N) \sum_{j=1}^{N} w_{\mathrm{DNN},j}. \tag{7}$$

$w_{\mathrm{DNN}}$ is the weight assigned to pre-trained DNN by the hierarchical assignment module.

### 2.4 Training procedure

The training procedure of IME is overviewed in Algorithm 1. First, the model is trained in an end-to-end way to minimize the overall loss given in Eq. 1, until convergence. Then, the experts are frozen, and the assignment module is trained independently to minimize the prediction loss given in Eq. 2. This alternating optimization approach first promotes expert specialization, and then improves the assignment module accuracy on trained experts. Since experts and the assignment module might be based on different architectures, they might converge at different rates (Ismail et al., 2020b), thus, different learning rates are employed for them.

---

**Algorithm 1:** Train Interpretable Mixture of Experts

---

**Input:** Features $X$, targets $Y$, expert learning rate $\tau$, assignment learning rate $\rho$, hyperparameters $\beta, \gamma, \delta, \lambda$
**Initialize:** All experts parameters $f$, the assignment module $A_w$ & $A_y$. Also set previous error $E = 0$
**while** *not converged* **do**
> $\overline{X} = [X; E]$;
> $\mathcal{L} = \mathcal{L}_{pred}\left(f, A_w, X, \overline{X}, Y\right) + \beta\mathcal{L}_{util}\left(A_w, \overline{X}\right) + \gamma\mathcal{L}_{div}\left(f, X\right) + \delta\mathcal{L}_{smooth}\left(A_w, \overline{X}\right) + \lambda\mathcal{L}_{A_y}\left(A_y, \overline{X}, Y\right)$;
> $f = f - \tau\,\nabla L$;
> $A_w = A_w - \rho\,\nabla L$;
> $A_y = A_y - \rho\,\nabla L$;
> **Update** E;

**Freeze experts and train assignment module**;
**while** *not converged* **do**
> $\mathcal{L} = \mathcal{L}_{pred}\left(f, A_w, X, \overline{X}, Y\right)$;
> $A_w = A_w - \rho\,\nabla L$;

---

## 3 IME interpretability capabilities

### 3.1 S-IME$_i$

Each prediction can be expressed as a switch statement with the cases specifying the prediction functions of experts, and the case conditions specifying the assignment predicate corresponding to each expert. Thus, we effectively have a single interpretable function defining each prediction as a *globally*-interpretable model. This can be useful for regulation-sensitive applications where the exact input-output relationships are needed, such as criminal justice systems.

### 3.2 S-IME$_d$

When the assignment isn't interpretable, the stability property of IME allows the data to be clustered into $n$ subsets such that the prediction in each subset comes from a single interpretable expert. This can be likened to *local* interpretability, as a separate interpretable function expresses the prediction in each cluster. Post-hoc interpretability methods can be used to assess feature importance for different assignments. This can be used for model debugging to verify that each expert's logic is correct.

### 3.3 H-IME$_i$ & H-IME$_d$

For hierarchical assignment, the DNN assignment module decides whether a sample is *easy* i.e. it can be accurately predicted by simple interpretable models vs. *difficult* i.e. it requires complex models, this is shown through additional experiments in the Appendix C.6. Similar to S-IME$_d$, one can obtain local interpretations for the *easy* samples. If explainability for the *difficult* samples is desired, post-hoc interpretability methods such as (Lundberg & Lee, 2017) may be adapted, however, their fidelity and faithfulness would be worse than the explanations coming from the inherently-interpretable experts. In addition, understanding why particular samples are assigned as *easy* vs. *difficult* can give insights into different data distribution modes (e.g. different seasonal climates and clothing sales), significant regime changes over time (e.g. after an ad campaign is launched for a product), and data anomalies (e.g. when a pandemic outbreak occurs).

**Accuracy-interpretability trade-off with H-IME.** The main difference between H-IME and S-IME is using a pre-trained DNN as an expert. The number of samples that are assigned to pre-trained DNN expert can be controlled by $\alpha U_{\text{DNN}}$ as described in Eq. 7. Increasing $\alpha$ yields less samples being assigned to the DNN expert, and hence constitutes a mechanism to increase the ratio of samples for which we use interpretable decision making. The effect of changing the value of $\alpha$ is empirically shown in Sec. 4.1. For some real-world datasets, H-IME can be highly valuable in preserving the accuracy (better than fully-interpretable S-IME) while utilizing interpretable models for a significant majority of samples.

## 4 Experiments

We evaluate the performance of IME on numerous real-world tabular and time-series datasets. As tabular data, we examine both the ones with time component, where the assignment module takes past errors as input, and the ones without past errors. Detailed descriptions of datasets and hyperparameter tuning are available in the Appendix C.

### 4.1 Tabular tasks with time component

We conduct experiments on the Rossmann Store Sales data (Rossmann), which has 30 features. Note that for this dataset, there is time feature, but we do not have sequential encoding of features from many timesteps, thus the problem is treated as tabular data prediction task.

**Baselines.** We compare IME to black-box models including Multi-layer Perceptron (MLP) (Gardner & Dorling, 1998), CatBoost (Dorogush et al., 2018), LightGBM (Ke et al., 2017) and XGBoost (Chen & Guestrin, 2016); and Inherently-interpretable Neural Additive models (NAM) (Agarwal et al., 2020), linear regression (LR) and shallow DTs. Our goal is to analyze the achievable performance and the accuracy vs. interpretability trade-off.

**Training and evaluation.** We use $n=20$ experts, either LR, shallow soft DTs (Frosst & Hinton, 2017), or a mixture of both. For interpretable assignment, a LR is used as the assignment module.

**Results.** Table 1 shows the performance of baselines and IME in RMSE. We observe a significant performance gap between a single interpretable model and a black-box model like MLP. Using S-IME$_i$ with an interpretable assignment module (the first two rows in the IME section in Table 1), we observe significant outperformance compared to a single interpretable model, but underperformance compared to black-box models – as expected. The performance becomes comparable to black-box models with S-IME$_d$, using a DNN assignment module. This underlines the importance of high capacity assignment for some problems. IME with hierarchical-level assignment, H-IME$_i$ and H-IME$_d$, yields the the best accuracy, while offering partial interpretability, as shown in Fig. 3, and is achieved by changing the penalty parameter for DNN assignment. As expected, the more samples are assigned to interpretable experts, the less accurate the model becomes. We observe that 20% of samples can be assigned to interpretable models with LR experts and 40% with soft DT experts, with almost no loss in accuracy – a large portion of the data is *sufficiently easy* to be captured by interpretable models.

| Baselines | Model Name | RMSE | IME | Assigner | Expert | Expert type | Interp. samples | RMSE |
|---|---|---|---|---|---|---|---|---|
| | MLP | 457.72 | S-IME$_i$ | Interp. | Interp. | Linear | 100.00 % | 1298.57 ± 96.95 |
| Black-box | CatBoost | 520.07 | S-IME$_i$ | Interp. | Interp. | SDT | 100.00 % | 820.83 ± 84.50 |
| Model | LightGBM | 490.57 | S-IME$_d$ | DNN | Interp. | Linear | 100.00 % | 671.46 ± 94.34 |
| | XGBoost | 567.8 | S-IME$_d$ | DNN | Interp. | SDT | 100.00% | 547.72 ± 43.06 |
| | | | S-IME$_d$ | DNN | Interp. | Linear/SDT | 100.00 % | 565.20 ±91.43 |
| | | | H-IME$_i$ | Interp. | Interp./DNN | DNN/Linear | 18.10% | 453.33 ± 37.53 |
| Interpretable | Linear | 1499.45 | H-IME$_i$ | Interp. | Interp./DNN | DNN/SDT | 40.01% | 486.94 ± 17.84 |
| Model | Soft DTs (SDT) | 1181.17 | H-IME$_d$ | DNN | Interp./DNN | DNN/Linear | 32.00% | 506.53 ± 11.93 |
| | NAM | 1497.47 | H-IME$_d$ | DNN | Interp./DNN | DNN/SDT | 43.32% | **451.93** ±14.08 |

**Table 1:** Performance of different methods on Rossmann dataset in terms of root mean-squared error (RMSE) . "IME with interpretable assignment module" performs better than a single interpretable model but worse than black box models. "IME with DNN assignment module" performance is comparable with that of black box models. "Hierarchical IME with a DNN expert" outperforms the best model while offering partial interpretability.

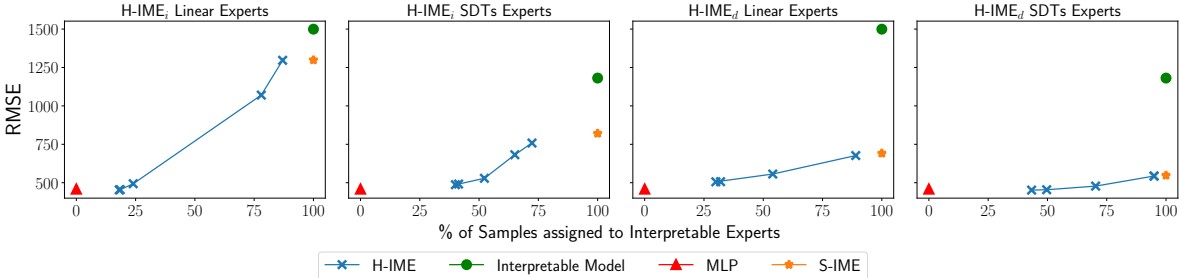

**Figure 3:** Partial interpretability by H-IME$_i$ and H-IME$_d$. Different accuracy values are obtained by sweeping the DNN penalty hyperparameter $\alpha$. With $\sim 40\%$ of samples assigned to soft DT experts, almost no degradation in accuracy is observed.

## 4.2 Tabular tasks without time component

On standard tabular tasks without time component, we compare the performance of S-IME$_i$ (interpretable assignment and interpretable experts) with other interpretable models. For classification, Telecom Churn Prediction (tel, 2018), Breast Cancer Prediction (Dua & Graff, 2017) and Credit Fraud Detection (Dal Pozzolo, 2015) datasets; while for regression, FICO Score Prediction (fic, 2018) datasets are used.

**Baselines.** For classification tasks, we compare with other inherently-interpretable such as Neural Additive models (NAMs) (Agarwal et al., 2020), Explainable Boosting Machines (EBMs) (Caruana et al., 2015), Black box models such as Deep Neural Networks (DNNs) and Gradient Boosted Trees (XGBoost).

**Results.** For classification tasks, we observe the outperformance of S-IME$_i$ compared to other inherently-interpretable and black-box models, as shown in Table 2. For regression tasks, none of the inherently-interpretable models outperforms DNNs on the FICO dataset, however, S-IME$_i$ is observed to be the second best. Results for partially interpretable versions of IME are provided in the Appendix C.3.

| Models | | AUC | | | RMSE |
|---|---|---|---|---|---|
| | | Telecom Churn | Breast Cancer | Credit | FICO |
| Simple Interpretable | Regression | .849 | .995 | .975 | 4.344 |
| | DTs | .824 | .992 | .956 | 4.900 |
| Inherently Interpretable | NAMs | - | - | *.980* | 3.490 |
| | EBMs | *.852* | .995 | .976 | 3.512 |
| | S-IME$_i$ | **.860** | **.999** | *.980* | *3.370* |
| Non-Interpretable (Black-Box) | XGBoost | .828 | .992 | **.981** | 3.345 |
| | DNNs | .851 | *.997* | *.980* | **3.324** |

**Table 2:** Higher AUCs and lower RMSEs are better. **Bold** indicates best results, while *italic* is second best. We report results on a regression dataset (FICO) for understanding credit score predictions, as well as three classification datasets: Credit, Telecom Churn, and Breast Cancer. Baseline results are from (Caruana et al., 2015) and (Interpretml, 2020).

## 4.3 Time-series forecasting tasks

We conduct experiments on multiple real-world time-series datasets for forecasting tasks, including Electricity (Electricity), Climate (Climate) and ETT (Zhou et al., 2021). Unlike tabular data, time-series datasets have models that encode information from multiple timesteps for each prediction.

**Baselines.** We compare to DNNs including LSTM (Hochreiter & Schmidhuber, 1997), Transformer (Vaswani et al., 2017), TCN (Lea et al., 2017) and Informer (Zhou et al., 2021); and interpretable models including LR and autoregressive (AR) models (Triebe et al., 2019).

**Training and evaluation.** We use $n$=10 LR experts for IME and LR or LSTM assignment modules for S-IME$_i$ and S-IME$_d$. We conduct hyperparameter tuning using grid search based on the validation performance.

| Features | Datasets | Forecast horizon | Black Box Models | | | | White Box Models | | IME | |
|---|---|---|---|---|---|---|---|---|---|---|
| | | | LSTM | Informer | Transformer | TCN | AR | LR | S-IME$_i$ | S-IME$_d$ |
| Univariate | Electricity | 24 | .178 | .159 | .163 | .172 | .173 | .167 | **.158** | **.158** |
| | | 48 | .204 | .188 | .199 | .195 | .198 | .188 | **.180** | **.180** |
| | | 168 | .251 | .234 | .252 | .235 | .242 | .224 | **.217** | .224 |
| | Climate | 24 | .094 | .106 | .094 | .092 | .095 | .098 | **.090** | .091 |
| | | 48 | **.128** | .176 | .131 | .134 | .140 | .141 | .137 | .140 |
| | | 168 | .230 | .313 | .242 | **.209** | .228 | .234 | .222 | . 220 |
| | ETTh1 | 24 | .085 | .075 | .077 | .084 | .030 | .034 | **.027** | **.027** |
| | | 48 | .202 | .109 | .131 | .133 | .049 | .054 | **.040** | .042 |
| | | 168 | .293 | .228 | .140 | .225 | .089 | .109 | **.071** | **.071** |
| | ETTh2 | 24 | .093 | .121 | .073 | .076 | .067 | .071 | **.065** | .066 |
| | | 48 | .122 | .145 | .108 | .110 | .104 | .101 | **.096** | **.096** |
| | | 168 | .256 | .253 | .169 | .248 | .176 | .173 | **.167** | **.167** |
| | ETTm1 | 24 | .017 | .018 | .017 | .017 | .013 | **.011** | .012 | **.011** |
| | | 48 | .029 | .056 | .036 | .032 | **.020** | **.020** | .021 | **.020** |
| | | 168 | .161 | .171 | .137 | .110 | .060 | .049 | .047 | **.044** |
| Multivariate | Climate | 24 | **.066** | .088 | .092 | .067 | .127 | .079 | .072 | .072 |
| | | 48 | .098 | .119 | .216 | **.095** | .171 | .106 | .097 | .097 |
| | | 168 | .204 | .220 | .252 | **.188** | .285 | .197 | .192 | .191 |
| | ETTh1 | 24 | .112 | .282 | .138 | .129 | .392 | .051 | **.047** | **.047** |
| | | 48 | .301 | .694 | .274 | .148 | 1.167 | .085 | .072 | **.070** |
| | | 168 | .518 | 1.027 | .303 | .172 | 1.788 | .142 | **.124** | .144 |
| | ETTh2 | 24 | .224 | .385 | .346 | .263 | .244 | .105 | .098 | **.091** |
| | | 48 | .590 | 1.557 | .582 | .772 | .738 | .390 | **.260** | .263 |
| | | 168 | .923 | 2.110 | 1.124 | .817 | .770 | .578 | **.477** | .548 |
| | ETTm1 | 24 | .034 | .070 | .029 | .035 | .024 | .022 | **.017** | **.017** |
| | | 48 | .065 | .109 | .074 | .038 | .067 | .031 | **.029** | **.029** |
| | | 168 | .242 | .371 | .430 | .127 | .929 | **.069** | .078 | .078 |
| | Winning counts | | 2 | 0 | 0 | 3 | 1 | 2 | 16 | 14 |

**Table 3:** The MSE of baselines and S-IME for various time-series datasets at different forecasting horizons. Note that here S-IME already outperforms DNNs H-IME is not considered.

**Results.** Table 3 shows the MSE for different datasets and forecasting horizons. For univariate forecasting, IME *outperforms* black-box models, while for multivariate forecasting, IME is comparable with black-box models (second-best accuracy after TCN). We also observe that using an interpretable assignment doesn't degrade the accuracy.

### 4.4 Interpretability analyses

**Global interpretability.** In S-IME$_i$, with both the assignment module and experts being interpretable, explanations are reduced to concise formulations. Algorithm 2 exemplifies the discovered interpretable model for *Electricity* dataset with input sequence length $t = 3$ and the number of experts $n = 2$, with $e$ denoting the expert errors. Note that when comparing global interpretability offered by IME with that of a single interpretable model, IME explanations might be harder to interpret since they involve an $n$-way classification followed by the explanation produced by each expert.

---

**Algorithm 2:** The interpretable model discovered by S-IME$_i$

$A_{expert_1} = -.05x_{t-1} + .03x_{t-2} + .9x_{t-3} + .38e_1 - .06e_2$
$A_{expert_2} = -.03x_{t-1} + .11x_{t-2} + .08x_{t-3} + .23e_1 + .05e_2$
**if** $A_{expert_1} > A_{expert_2}$ **then**
| $y_{t+1} = -.049x_{t-1} - .29x_{t-2} + 1.17x_{t-3}$
**else**
| $y_{t+1} = -.081x_{t-1} - .23x_{t-2} + 1.13x_{t-3}$

---

**Expert interpretability.** All IME options provide local expert interpretability, in which case, explanations can be given as equations for each expert, or can be conveniently visualized. Fig. 4 exemplifies this on Rossmann.

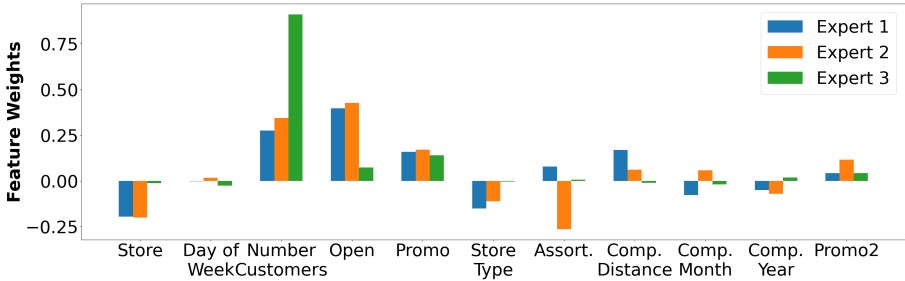

**Figure 4:** Expert interpretability. Normalized weights for different LR experts on Rossmann, allowing expert behavior to be easily interpreted. "Expert 1" puts high positive weights on 'the number of customers', 'open' and 'competition distance'; "Expert 2" on 'assortment'; and "Expert 3" on 'the number of customers'.

**Sample-wise expert interpretability.** Fig. 5 shows feature weights of the interpretable LR model for different samples on Rossmann. For the first sample shown in Fig. 5a, the most influential feature is the number of customers entering the store. Whereas for the second sample shown in Fig. 5b, multiple features influence the model output.

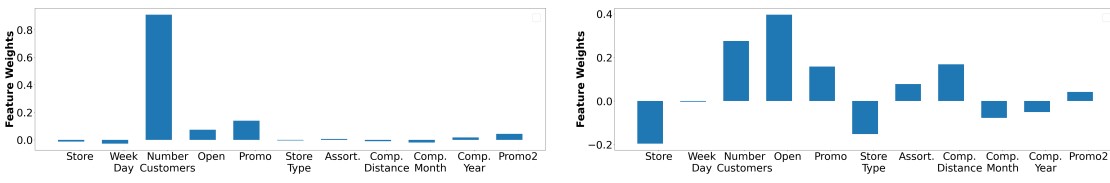

(a) The most influential feature is the number of customers.      (b) Multiple features influence the model output.

**Figure 5:** Sample-wise feature weights on Rossmann dataset.

**Identifying data distribution modes.** We construct a synthetic dataset (Fig. 6a) to showcase the additional interpretability capabilities. We run S-IME$_i$ with LR assignment (Fig. 1a) and S-IME$_d$ with MLP assignment (Fig. 1b). Fig. 6b shows that both S-IME$_i$ and S-IME$_d$ assign most samples so that $Y = x_1$ are assigned to Expert 1, $Y = x_2$ to Expert 2 and $Y = x_3$ to Expert 3. In this way, IME yields insights into how different subsets can be split based on unique characteristics.

**Identifying incorrect model behavior.** Another use case for interpretability is model debugging and identifying undesired behaviours. To showcase IME for this, we focus on synthetic data shown in Fig. 6a. Fig. 6c shows the weights for the experts for S-IME with interpretable (top) and DNN assignment (bottom) respectively. S-IME$_i$ yields almost ideal behavior; Expert 1 assigns highest weight to feature $x_1$, Expert 2 to $x_2$ and Expert 3 to $x_3$. However, Expert 1 in S-IME$_d$ incorrectly assigns the highest weights to feature $x_2$. Investigating different weights in this way can help verify whether the expert logic is correct. Model builders can benefit from this insights to debug and improve model performance, e.g. by replacing or down-weighing certain experts.

**Identifying temporal regime changes.** Fig. 7a shows a synthetic univariate dataset where the feature distribution changes over time. S-IME$_i$ with LR assignment and S-IME$_d$ with an LSTM assignment module are used, and all interpretable experts are simple auto-regressive models. Fig. 7b and 7c show that IME can

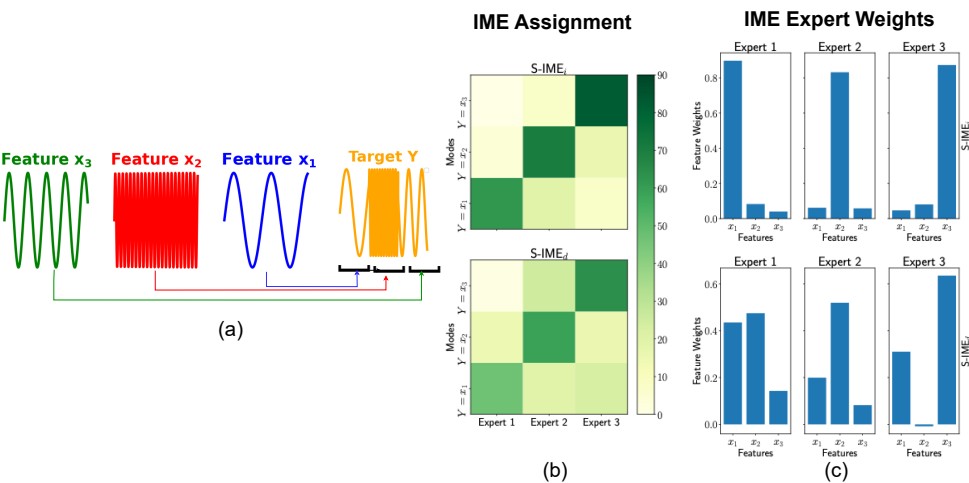

**Figure 6:** Identifying different clusters for S-IME$_i$ and S-IME$_d$. (a) Synthetic data with three features. (b) Number of samples from different modes assigned to each expert. (c) Weights given to each feature by different experts.

identify changes over time, using different experts for distribution modes. This capability can be used to get insights into temporal characteristics and major events.

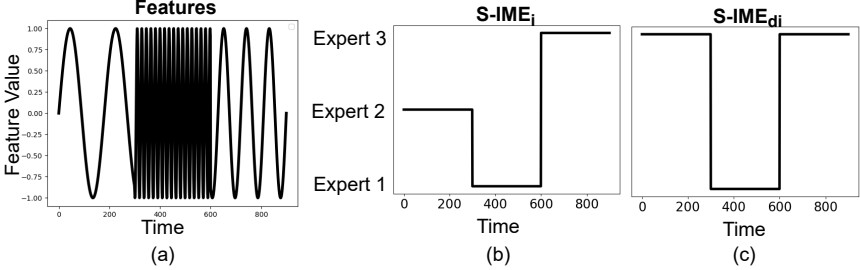

**Figure 7:** Identifying regime changes. (a) Synthetic univariate time-series where feature distribution changing over time. (b) S-IME$_i$ identifies changes in time regime and utilize all 3 experts. (c) S-IME$_d$ identifies changes in regime but only 2 experts are used.

**User study on IME explanations.** IME offers faithful local explanations, as the actual interpretable models behind each prediction are known. This is in contrast to post-hoc methods used to explain black-box methods such as SmoothGrad (Smilkov et al., 2017), SHAP (Lundberg & Lee, 2017), Integrated Gradients (Sundararajan et al., 2017), and DeepLift (Shrikumar et al., 2017). Such methods may be unreliable (Adebayo et al., 2018; Hooker et al., 2019; Ghorbani et al., 2019), especially for time series (Ismail et al., 2020a). To demonstrate the quality of IME's explanations, we design a user study that focuses on comparisons with the commonly-used post-hoc method, SHAP (Lundberg & Lee, 2017) (we are referring to the quality of the explanations from a user perspective i.e., given an explanation will the user be able to understand, interact and modify a model based on the explanation?). We base the objective component on human-grounded metrics (Doshi-Velez & Kim, 2018), where the tasks conducted by users are simplified versions of the original task. We use the task of sales prediction on Rossmann, and experiment with S-IME$_i$ (LR assignment & 20 experts) and MLP as the black-box model. First, we consider a counterfactual simulation scenario. For each sample, the participants are given an input and an explanation (along with access to training data for analyses) and asked how the model output (sales prediction) would change with the input changes. The participant are able to choose one of the options: no change, increase or decrease. The same sample with explanations from the chosen interpretable experts of IME and SHAP (Lundberg & Lee, 2017) are provided (as shown in Appendix Fig 12 & Fig 13. All examples shown are correctly classified by both models. In total, we collect 77 samples from 15 participants.

When provided with IME explanations, participants can predict model behavior with an accuracy of 69% vs. 42% of SHAP. This shows that explanations provided by IME can be easily understood by participants and are more faithful. Next, we also ask the participants which explanations they trust more for each sample: explanation A/B, both, or neither. IME is chosen for 87% of the cases vs. 6.5% of SHAP (and neither gets 6.5%), demonstrating the trustworthiness of IME's explanations. The Appendix D.1 includes more details.

## 4.5 Ablation studies

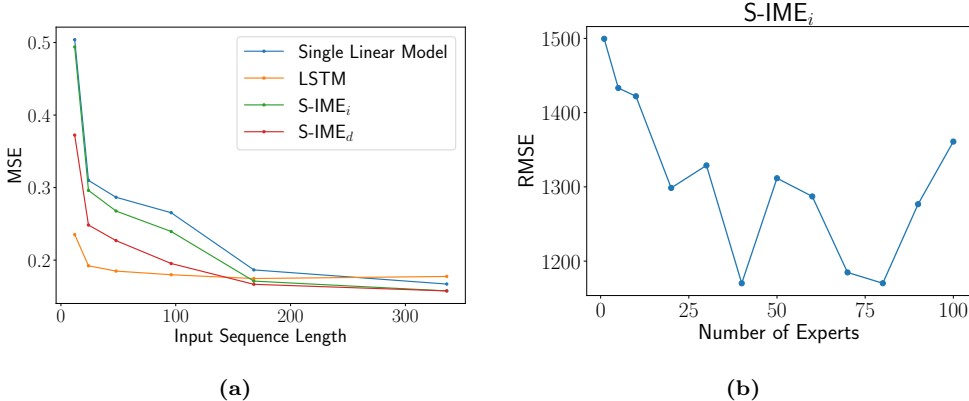

**Figure 8:** **(a)** Error with increased model capacity for S-IME with LR on Electricity. **(b)** Number of experts vs. accuracy on Rossmann.

**Comparison to black-box model performance.** Table 3 shows that IME can outperform black-box models. To further shed light on this, we investigate the effect of changing the interpretable model's capacity. For LR experts, as we increase the input sequence length (and hence the number of learnable coefficients), the model accuracy increases as shown in Fig. 8a. Note that increasing the sequence length doesn't affect the number of parameters for an LSTM. IME outperforms a single interpretable model for any sequence length, and starts outperforming LSTM when the accuracy gap between LR and LSTM gets smaller.

**Effect of the number of experts.** Fig. 8b shows the effect of the number of experts. As the number of experts increases, the accuracy increases until a certain optimal (40 for LR experts, 20 for soft DTs). After this, increasing the number of experts causes a slight decrease in accuracy as experts become underutilized. The number of experts can be treated as a hyperparameter and optimized on a validation dataset. We also note that IME with fewer experts can be desirable for improving overall model interpretability.

**Contributions of IME components.** We perform ablation studies on important IME components, shown in Table 4. Removing each yields worse performance. Note that IME can be used for *any tabular dataset* without a time component by removing the past error from the input to the assignment module, i.e $A([X_t; E_{t-1}]) \rightarrow A(X)$ (as shown in Sec. 4.2). However, the availability of errors over time improves the performance of the assignment module.

## 5 Related Work

**Mixture of experts.** Jacobs et al. introduced ME over three decades ago. Since then many expert architectures have been proposed such as SVMs (Collobert et al., 2002) , Dirichlet Processes (Shahbaba & Neal, 2009) and Gaussian Processes (Tresp, 2001). (Jordan & Jacobs, 1994) propose hierarchical assignment for MEs. (Shazeer et al., 2017) propose an effective deep learning method that stacked ME as a layer between LSTM layers. (Shazeer et al., 2018; Lepikhin et al., 2020; Fedus et al., 2021) incorporate ME as a layer in Transformers. (Pradier et al., 2021) introduce human-ML ME where the assignment module depends on human-based rules, and the experts themselves are black-box DNNs. IME combines interpretable experts with DNNs to produce inherently-interpretable architectures with similar accuracy, comparable to DNNs.

| Loss function | S-IME$_i$ | S-IME$_d$ |
|---|---|---|
| Proposed IME | 1298.57 | 671.46 |
| Without $\mathcal{L}_{util}$ | 1369.54 | 710.61 |
| Without $\mathcal{L}_{smooth}$ | 1481.87 | 844.15 |
| Without $\mathcal{L}_{A_y}$ | 1342.89 | 701.50 |
| Without using past errors | 1338.57 | 745.23 |
| Without freezing step (Sec. 2.4) | 1315.70 | 760.25 |

**Table 4:** Ablation studies on Rossmann (metrics in RSME).

**Interpretable DNNs for structured data.** (Agarwal et al., 2020) propose NAM which uses a DNN per feature. NAM does not consider feature-feature interactions and thus is not suitable for high-dimensional data. To address this, NODE-GAM (Chang et al., 2021) is introduced, modifying NODE (Popov et al., 2019) into a generalized additive model. Both NAM and NODE-GAM are only applicable to tabular data. (Luo et al., 2021; Irsoy et al., 2012; Frosst & Hinton, 2017) propose the use of a soft DT to make DNNs more interpretable. N-Beats (Oreshkin et al., 2019) uses residual stacks to constrain trend and seasonality functional forms to generate interpretable stacks for univariate time series. (Shulman & Wolf, 2020) create a per-user decision tree for tabular recommendation systems. In contrast to these, IME (1) provides explanations that accurately describe the overall prediction process with minimal loss in accuracy, (2) supports both tabular and time-series data, and (3) can be easily used for complex large-scale real-world datasets.

## 6 Conclusions

We propose IME, a novel inherently-interpretable framework. IME offers different options, from providing explanations that are the exact description of how prediction is computed via interpretable models, to adjusting what ratio of samples can be predicted with interpretable models. On real-world tabular and time-series data, while achieving useful interpretability capabilities (demonstrated on synthetic datasets as well as with user studies), the accuracy of IME is on par with, and in some cases even better than, state-of-the-art DNNs. We leave extension of IME to unstructured high-dimensional data types, such as image and text, to future work.

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

## A   H-IME Framework

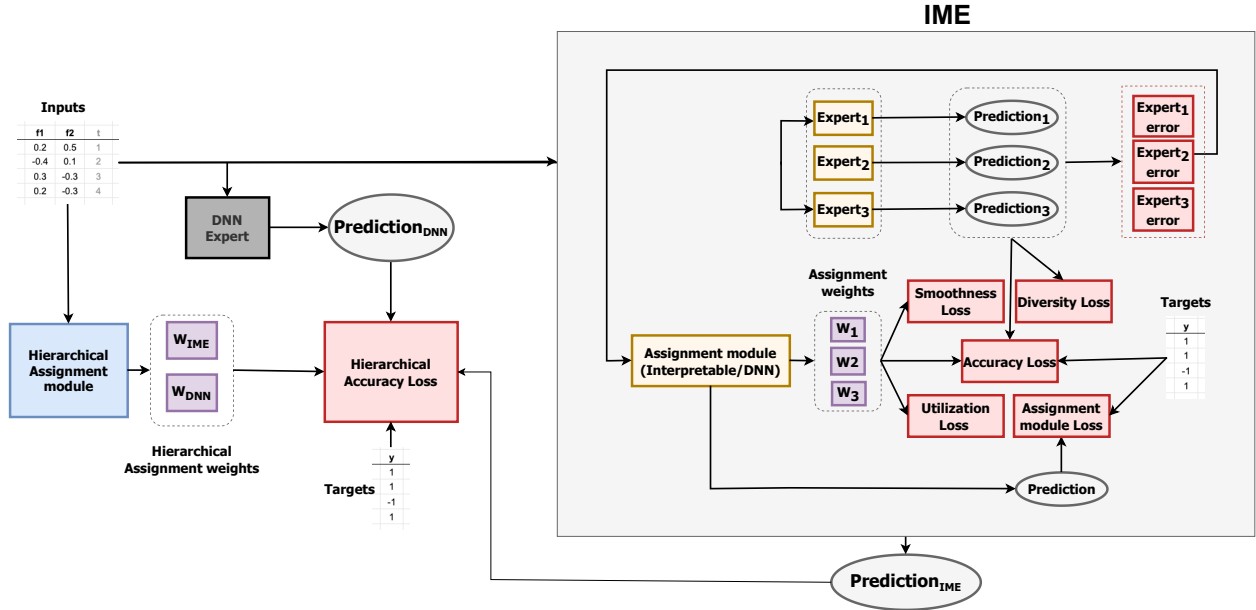

**Figure 9:** H-IME consists with DNN expert and IME.

## B   IME for classification

To adapt IME for classification problems, the model outputs are the output class probabilities instead of regression value. Correspondingly, the prediction loss $\mathcal{L}_{pred}$ is changed from Eq. 2 to the following:

$$\mathcal{L}_{pred}(f, A_w, X, \overline{X}, Y) = -\log \sum_{i=1}^{n} A_w(\overline{X})_i \frac{e^{-CE(Y, f_i(X))/2}}{\sqrt{2\pi}}, \tag{8}$$

where $CE(Y, f_i(X))$ is standard softmax cross entropy.

## C  Experimental details

All experiments were ran on a NVIDIA Tesla V100 GPU. We perform time-series data experiments in two settings – multivariate and univariate forecasting. The multivariate setting involves multivariate inputs and a single output. The univariate setting involves univariate inputs and outputs, which are the target values described below. We use MSE as the evaluation metric.

### C.1  Datasets

We perform a 70/10/20 train/validation/test split for each dataset.

**Tabular tasks without time component**

- **Telecom Churn Prediction [Classification]** Customer churn is the rate at which customers are lost, used by telecom companies to predict the number of customers that will leave a telecom service provider. Churn dataset (tel, 2018) consists of features such as services booked, account information, and demographics overall there are 20 features and 7043 samples. The task is to classify a customer as churn or not churn.
- **Breast Cancer Prediction [Classification]** The dataset is based on ML Breast Cancer Wisconsin dataset (Dua & Graff, 2017), consisting 30 real-valued features that are computed for each cell nucleus and 569 samples. The goal is to predict if the tissue is malignant or benign.
- **Credit Fraud Detection [Classification]** The dataset (Dal Pozzolo, 2015) contains transactions made by credit cards. The dataset consists of 28 features obtained from PCA and 284,807 transactions. The task is to predict whether a given transaction is fraudulent or not. Note that this dataset is highly unbalanced with only 0.172% of samples labeled as fraud.
- **FICO Score Prediction [Regression]** The FICO score is a proprietary credit score to determine creditworthiness for loans in the United States. The dataset (fic, 2018) is comprised of real-world credit applications made by customers and their assigned FICO Score, based on their credit report information. The dataset consists of 23 features and the goal is to predict scores based on different features.

**Tabular tasks with time component**

We conduct experiments on the Rossmann Store Sales data (Rossmann), which has 30 features. Note that there is time feature, but we do not have sequential encoding of features from many timesteps, thus the problem is treated as tabular. The dataset consists of samples from 1115 stores. The goal is to predict daily product sales based on sales history and other factors, including promotions, competition, school and state holidays, seasonality, and locality.

**Time-series forecasting tasks**

- **Electricity**[1] The dataset measures the electricity consumption of 321 clients. We convert the dataset into hourly-level measurements and forecast the consumption of different clients over time.
- **Climate**[2] The dataset consists of 14 different quantities (air temperature, atmospheric pressure, humidity, wind direction, etc.), recorded every 10 minutes between 2009-2016. We convert the dataset into hourly-level measurements and forecast the hourly temperature over time.
- **ETT** We conduct experiments on ETT (Electricity Transformer Temperature) (Zhou et al., 2021). This consists three datasets: two hourly-level datasets (ETTh) and one 15-minute-level dataset (ETTm), measuring six power load features and "oil temperature", the chosen target value for univariate forecasting.

### C.2  Hyperparameters

All baselines were tuned using (Liaw et al., 2018) with at least 20 trials. Hyperparameter search grid used for each model is available in Table 5. To avoid overfitting, dropout and early stopping were used. The best hyperparameters were chosen based on the validation dataset while the results reported in the table

---

[1]https://archive.ics.uci.edu/ml/datasets
[2]https://www.kaggle.com/mnassrib/jena-climate

are on the test dataset. For all IME experiments we set $k = 1$ for $\mathcal{L}_{util}$. For $\mathcal{L}_{div}(f, X)$, we set $\tau = 0.2$. All models were trained using Adam optimizer, and the batch size and learning rate values were modified at each experiment.

**Rossmann** Hyperparameter used for IME are give in table 6. For baselines: *CatBoost, LightGBM* and *XGBoost*, we used parameters specified by benchmark [3]. *LR*: The batch size is 512 and learning rate is .001. *Soft DT (SDT)*: The batch size is 512, learning rate is .0001 and the depth is 5. *MLP*: The batch size is 512, 5 layers are used, each containing 128 hidden units with a learning rate .001. *NAM*: Each feature is modeled using a MLP with 2 hidden layer each with 32 hidden units, with a batch size 512 and learning rate .001.

**Electricity** For IME, we use a batch size of 512 and a sequence length 336; the remaining hyperparameters are available in Table 8. While hyperparameters for baseline are in Table 9.

**Climate** For IME, a batch size of 512 and a sequence length 336 was used the remaining hyperparameters are available in Table 10. While for baselines detailed hyperparameters are available in Table 11.

| | LSTM | TCN | Transformer | Informer |
|---|---|---|---|---|
| Learning rate | 0.00001 to 0.1 | 0.00001 to 0.01 | 0.00001 to 0.01 | 0.00001 to 0.01 |
| Batch size | 64,128,256 | 64,128,256 | 64,128,256 | 64,128,256 |
| Hidden units | 128, 256,512 | 128, 256,512 | 128, 256,512 | 2048 |
| Encoder layers | 2 to 6 | N/A | 2 to 6 | 1 to 3 |
| Decoder layers | N/A | N/A | 2 to 6 | 1 to 3 |
| Levels | N/A | 1 to 10 | N/A | N/A |
| Kernal | N/A | 5 to 15 | N/A | N/A |
| Attention heads | N/A | N/A | 2,4,8 | 8 |
| Embedding sizes | N/A | N/A | 128, 256, 512 | 512 |

**Table 5:** Hyperparameter search grid used for different models

| IME | Number of experts | Learning rates | | Model Hyperparameters | | | |
|---|---|---|---|---|---|---|---|
| | | $\tau$ | $\rho$ | $\beta$ | $\gamma$ | $\delta$ | $\lambda$ |
| Linear Assign. LR Expert | 20 | .0001 | .001 | 10 | 0 | .1 | 1 |
| Linear Assign. SDT Expert | 20 | .0001 | .001 | .1 | 0 | .1 | 1 |
| MLP Assign. LR Expert | 20 | .0001 | .001 | .1 | 0 | .1 | 1 |
| MLP Assign. SDT Expert | 20 | .0001 | .001 | .1 | 0 | .1 | 1 |

**Table 6:** IME Hyperparameters for Rossmann dataset.

| Dataset | Assigner | Expert | Number of experts | Learning rates | | Hyperparameters | | | |
|---|---|---|---|---|---|---|---|---|---|
| | | | | $\tau$ | $\rho$ | $\beta$ | $\gamma$ | $\delta$ | $\lambda$ |
| Telecom Churn | LR | LR | 2 | 0.001 | 0.0001 | 1 | 0 | 0 | 1 |
| Breast Cancer | LR | LR | 2 | 0.1 | 0.01 | 1 | 1 | 0 | 1 |
| Credit | LR | LR | 3 | 0.001 | 0.0001 | 1 | 0 | 0 | 1 |
| FICO | LR | SDT | 2 | 0.01 | 0.0005 | 1 | 20 | 0 | 1 |

**Table 7:** IME Hyperparameters for different tabular datasets.

---

[3]https://github.com/catboost/benchmarks/blob/master/kaggle/rossmann-store-sales/README.md

| IME | | Number of | | Learning rates | | | Model Hyperparameters | | | |
|---|---|---|---|---|---|---|---|---|---|---|
| | | experts | | $\tau$ | $\rho$ | | $\beta$ | $\gamma$ | $\delta$ | $\lambda$ |
| Linear Assigner LR Expert | | 10 | | .0001 | .001 | | 10 | 0 | .1 | 1 |
| LSTM Assigner LR Expert | | 10 | | .0001 | .001 | | 1 | 0 | .1 | 1 |

**Table 8:** IME Hyperparameters for Electricity dataset.

| Model | Forecast horizon | Batch size | Sequence length | Learning rate | | | |
|---|---|---|---|---|---|---|---|
| Auto-regressive | 24 | 512 | 336 | .0001 | | | |
| | 48 | 512 | 336 | .0001 | | | |
| | 168 | 512 | 336 | .0001 | | | |
| LR | 24 | 512 | 336 | .0001 | | | |
| | 48 | 256 | 336 | .0001 | | | |
| | 168 | 512 | 336 | .0001 | | | |
| | Forecast Horizon | Batch Size | Sequence Length | Learning Rate | Hidden Units | Layers | |
| LSTM | 24 | 512 | 168 | .001 | 256 | 2 | |
| | 48 | 256 | 96 | .01 | 128 | 5 | |
| | 168 | 512 | 168 | .001 | 256 | 2 | |
| | Forecast Horizon | Batch Size | Sequence Length | Learning Rate | Hidden Units | Kernel | Level |
| TCN | 24 | 256 | 96 | .0001 | 512 | 8 | 8 |
| | 48 | 256 | 96 | .0001 | 512 | 8 | 8 |
| | 168 | 256 | 96 | .0001 | 512 | 8 | 8 |
| | Forecast Horizon | Batch size | Sequence length | Learning rate | Hidden units | Encoder | Decoder | Heads |
| Transformer | 24 | 512 | 168 | .001 | 256 | 3 | 5 | 4 |
| | 48 | 256 | 168 | .01 | 256 | 3 | 5 | 4 |
| | 168 | 512 | 168 | .001 | 256 | 3 | 5 | 4 |
| Informer | 24 | 512 | 168 | .0001 | 512 | 2 | 1 | 8 |
| | 48 | 512 | 168 | .0001 | 512 | 2 | 1 | 8 |
| | 168 | 512 | 168 | .0001 | 512 | 2 | 1 | 8 |

**Table 9:** Baseline Hyperparameters for Electricity dataset.

| Features | | IME | | Number of | | Learning rates | | | Model Hyperparameters | | | |
|---|---|---|---|---|---|---|---|---|---|---|---|---|
| | | | | Experts | | $\tau$ | $\rho$ | | $\beta$ | $\gamma$ | $\delta$ | $\lambda$ |
| Univariate | | Linear Assigner LR Expert | | 2 | | .002 | .005 | | 1 | 0 | .1 | 1 |
| | | LSTM Assigner LR Expert | | 2 | | .001 | .01 | | 10 | 0 | .1 | 1 |
| Multivariate | | Linear Assigner LR Expert | | 10 | | .0001 | .001 | | 1 | .001 | 10 | 1 |
| | | LSTM Assigner LR Expert | | 10 | | .0001 | .001 | | 1 | 0 | 10 | 1 |

**Table 10:** IME Hyperparameters for Climate dataset.

| | | | | Univariate | | | |
|---|---|---|---|---|---|---|---|
| Model | Forecast horizon | Batch size | Sequence length | Learning rate | | | |
| Auto-regressive | 24 | 256 | 336 | .001 | | | |
| | 48 | 256 | 336 | .005 | | | |
| | 168 | 256 | 336 | .005 | | | |
| LR | 24 | 256 | 336 | .005 | | | |
| | 48 | 256 | 336 | .001 | | | |
| | 168 | 256 | 336 | .005 | | | |
| | Forecast horizon | Batch size | Sequence length | Learning rate | Hidden units | Layers | |
| LSTM | 24 | 1024 | 24 | .01 | 128 | 4 | |
| | 48 | 256 | 240 | .001 | 128 | 5 | |
| | 168 | 1024 | 24 | .01 | 128 | 4 | |
| | Forecast Horizon | Batch Size | Sequence Length | Learning Rate | Hidden Units | Kernel | Level |
| TCN | 24 | 1024 | 96 | .001 | 512 | 4 | 3 |
| | 48 | 1024 | 96 | .001 | 512 | 4 | 3 |
| | 168 | 256 | 96 | .0001 | 512 | 8 | 8 |
| | Forecast Horizon | Batch Size | Sequence Length | Learning Rate | Hidden Units | Encoder | Decoder | Heads |
| Transformer | 24 | 64 | 168 | .0005 | 256 | 3 | 3 | 4 |
| | 48 | 64 | 96 | .0001 | 512 | 2 | 4 | 8 |
| | 168 | 128 | 336 | .0005 | 256 | 4 | 4 | 8 |
| Informer | 24 | 512 | 720 | .0001 | 512 | 2 | 1 | 8 |
| | 48 | 512 | 720 | .0001 | 512 | 2 | 1 | 8 |
| | 168 | 512 | 720 | .0001 | 512 | 2 | 1 | 8 |
| | | | | Multivariate | | | |
| Model | Forecast horizon | Batch size | Sequence length | Learning rate | | | |
| Auto-regressive | 24 | 256 | 168 | .0005 | | | |
| | 48 | 256 | 168 | .0005 | | | |
| | 168 | 256 | 336 | .001 | | | |
| LR | 24 | 256 | 96 | .001 | | | |
| | 48 | 256 | 96 | .001 | | | |
| | 168 | 256 | 168 | .0005 | | | |
| | Forecast Horizon | Batch Size | Sequence Length | Learning Rate | Hidden Units | Layers | |
| LSTM | 24 | 512 | 168 | .001 | 256 | 2 | |
| | 48 | 512 | 168 | .001 | 128 | 2 | |
| | 168 | 512 | 168 | .001 | 256 | 2 | |
| | Forecast horizon | Batch size | Sequence length | Learning rate | Hidden units | Kernel | Level |
| TCN | 24 | 1024 | 96 | .001 | 256 | 12 | 1 |
| | 48 | 1024 | 96 | .001 | 256 | 13 | 1 |
| | 168 | 1024 | 96 | .001 | 256 | 13 | 1 |
| | Forecast horizon | Batch size | Sequence Length | Learning rate | Hidden units | Encoder | Decoder | Heads |
| Transformer | 24 | 128 | 96 | .0005 | 256 | 5 | 4 | 8 |
| | 48 | 128 | 336 | .0005 | 128 | 4 | 4 | 8 |
| | 168 | 128 | 168 | .00005 | 256 | 2 | 3 | 4 |
| Informer | 24 | 512 | 168 | .0001 | 512 | 3 | 2 | 8 |
| | 48 | 512 | 96 | .0001 | 512 | 2 | 1 | 8 |
| | 168 | 512 | 336 | .0001 | 512 | 3 | 2 | 8 |

**Table 11:** Hyperparameters for Climate dataset.

### C.3 Additional regression experiments

For regression tasks on FICO dataset, we preformed partial interpretable variations of IMEs including S-IME$_d$, H-IME$_i$ and H-IME$_d$

**Regression Results:** From Table 12, we observe that for none of the inherently-interpretable models outperforms DNNs on FICO dataset, however, we partial interpretable models H-IME$_i$ can outperform DNN, with only 0.2% of samples assigned to the DNN.

| Models | | RMSE (FICO dataset) |
|---|---|---|
| Simple Interpretable | Regression | 4.344 |
| | DTs | 4.900 |
| Inherently Interpretable | NAMs | 3.490 |
| | EBMs | 3.512 |
| | S-IME$_i$ | *3.370* |
| Partial Interpretable | S-IME$_d$ | 3.380 |
| | H-IME$_i$ | **3.290** |
| | H-IME$_d$ | 3.360 |
| Non-Interpretable (Black-Box) | XGBoost | 3.345 |
| | DNNs | *3.324* |

**Table 12:** Lower RMSEs are better. **Bold** indicates best results, while *italic* is second best. We report results on a regression dataset (FICO) for understanding credit score predictions.

### C.4 DNNs using past errors

Past errors are considered as useful information for the IME assignment module as they can help the assignment module choose the best expert in the next time step. IME has a direct mechanism of benefiting from past errors – by yielding better assignment. One natural question can be whether the past errors would also benefit DNNs if they are simply input to the models for the downstream prediction task. To shed light onto this question, we experiment by using past errors as inputs to DNNs. The results on ETTm1 dataset are shown in Table 13. We find that adding past errors can help the accuracy of DNNs but even with past errors, the performance is significantly worse than that of IME. We also underline that IME does not use past errors for its experts while yielding downstream predictions, but only for its assignment module. We note that explainability with past errors is more natural for the assignment module, while less so for the predictive experts.

| Forecast horizon | Black Box Models | | | | Black Box Models + Past Errors | | | | IME | |
|---|---|---|---|---|---|---|---|---|---|---|
| | LSTM | Informer | Transformer | TCN | LSTM | Informer | Transformer | TCN | S-IME$_i$ | S-IME$_d$ |
| 24 | .017 | .018 | .017 | .017 | .015 | .015 | .017 | .016 | .012 | **.011** |
| 48 | .029 | .056 | .036 | .032 | .032 | .033 | .033 | .027 | .021 | **.020** |
| 168 | .161 | .171 | .137 | .110 | .113 | .142 | .123 | .168 | .047 | **.044** |

**Table 13:** The MSE of baselines, baselines with past errors and S-IME for univariate ETTm1 time-series dataset at different forecasting horizons.

### C.5 The impact of diversity loss

Multiple loss terms are employed in IME, as given in Eq. 1 and their contributions can be controlled via hyperparameters. In general, the benefit of some loss terms might be more prominent on some datasets. In

Table 4, we show the contribution of different IME components on the Rossmann dataset. For this particular dataset, $\mathcal{L}_{div}$ is observed to have a minor effect. We find that adding $\mathcal{L}_{div}$ has more effect on time series datasets where the output consists multiple forecasts rather than a single point prediction. In Table 14, we show the effect of $\mathcal{L}_{div}$ on the ETTm1 dataset – it is observed to help reduce MSE for different forecasting horizons.

| Forecast horizon | S-IME$_i$ | |
| --- | --- | --- |
| | Without $\mathcal{L}_{div}$ | With $\mathcal{L}_{div}$ |
| 24 | .014 | .012 |
| 48 | .023 | .021 |
| 168 | .055 | .047 |

**Table 14:** The MSE for S-IME$_i$ with and without $\mathcal{L}_{div}$ on univariate ETTm1 time-series dataset at different forecasting horizons.

### C.6 Hierarchical assignment differentiating between 'difficult' and 'easy' samples

H-IME assignments module can also identify 'difficult' samples (i.e., samples requiring a DNN for an accurate prediction) and 'easy' samples (i.e., samples that can be predicted by a simple interpretable model) offering insights on the task. To verify this we looked into the difference in accuracy between samples assigned to a DNN versus an interpretable expert for the Rossmann dataset. Table 1 showed that the best-performing model was H-IME where 43.32% of samples were assigned to an interpretable expert, in Table 15 we compare the RMSE of samples assigned to DNN with those assigned to SDT. We find that for samples assigned to DNN, DNN accuracy is much better than a single SDT this shows that these samples are in fact 'difficult' requiring more complex models to get an accurate prediction. For samples assigned to a single interpretable model, we find that a single SDT for those samples outperforms DNN, confirming that those samples are 'easy' samples where model complexity is not required for high performance rather it may harm overall accuracy.

| Sample Assignement | | DNN | SDT | S-IME | H-IME |
| --- | --- | --- | --- | --- | --- |
| DNN | i.e., difficult samples | 570.05 | 1292.12 | 842.27 | 518.97 |
| Interpretable expert (SDT) | i.e., easy samples | 120.87 | 92.1 | 97.64 | 23.54 |

**Table 15:** RMSE for different H-IME assignment on Rossmann.

### C.7 IME in comparison to other ME variants:

Sparsely-gated ME (Shazeer et al., 2017) assigns samples to a subset of experts and then combines the outputs of gates. Switch ME (Fedus et al., 2021) assigns each sample to a single expert. For fair comparison, a single expert is used at inference. Table 16 also shows that IME's assignment mechanism yields superior results.

| ME variant | RMSE |
| --- | --- |
| S-IME$_i$ | 1298.57 |
| Sparsely-gated ME (Shazeer et al., 2017) | 1575.18 |
| Switch ME (Fedus et al., 2021) | 2839.56 |

**Table 16:** IME in comparison to other ME variants on Rossmann (metrics in RSME).

# D   Interpretability Results

### D.1   User study details:

- The participants were given instructions shown in Fig. 10.
- Then, a data-sample from Rossmann was provided as shown in Fig. 11 and the participant s were asked to answer questions based on two explanations.
- Explanations from different models , and questions were shown to participant as shown in Fig. 12 and Fig. 13.
- Finally, the participant was asked which model he trusts more as shown in Fig. 14.

## Interpretability User Study Sample 1

The goal of this user study is to test two explanation methods generated from two different models. The task is to predict the model output given an explanation.

To test both explanations we will present examples from the Rossmann dataset. This is a retail dataset with the goal of predicting the daily sales of a store. Store sales are influenced by many factors, including promotions, competition, school and state holidays, seasonality, and locality. The dataset is available here
(https://docs.google.com/spreadsheets/d/1fmvh9UZLgEjpf3dQgiKz6Bzy8KEpEUEHIvGSOfpDcoQ/edit?usp=sharing)

Sales - the turnover for any given day (this is what you are predicting)

Store - a unique Id for each store
Customers - the number of customers on a given day
Open - an indicator for whether the store was open: 0 = closed, 1 = open
StateHoliday - indicates a state holiday. Normally all stores, with few exceptions, are closed on state holidays. Note that all schools are closed on public holidays and weekends. 1 = public holiday, 2 = Easter holiday, 3 = Christmas, 0 = None
School holiday - indicates if the (Store, Date) was affected by the closure of public schools
Store Type - differentiates between 4 different store models: 1, 2, 3, 4
Assortment - describes an assortment level: 1 = basic, 2 = extra, 3 = extended
Competition Distance - the distance in meters to the nearest competitor store
Competition Open Since [Month/Year] - gives the approximate year and month of the time the nearest competitor was opened
Promo - indicates whether a store is running a promo on that day
Promo2 - Promo2 is a continuing and consecutive promotion for some stores: 0 = store is not participating, 1 = store is participating
Promo2 Since [Year/Week] - describes the year and calendar week when the store started participating in Promo2
Promo2 [month] - describes if Promo2 was running in that month it takes a boolean value 0/1

**Use the training data to answer the following questions (you can try to find the nearest example and sort values to get the general trends on how input features affect the output).**

Based on the training dataset and your understanding of the problem please answer the questions in the following sections. In each section, you will be given a training example and previous input values (if any), then you will be asked a series of questions on that sample. In the first question, we ask you to guess the correct output given the sample. Then assume that you are given a model explanation and asked to guess the model output given multiple explanations.

**Figure 10:** User study instructions

Using the training data, predict the daily sales of the store in the sample below.

| Features | Time Step 1 |
|---|---|
| Store | 1 |
| Day Of Week | 1 |
| # Customers | 712 |
| Open | 1 |
| Promo | 1 |
| State Holiday | 0 |
| School Holiday | 1 |
| Year | 2015 |
| Month | 1 |
| Day | 5 |
| Store Type | 1 |
| Assortment | 1 |
| Competition Distance | 570 |
| Competition Open Since Month | 11 |
| Competition Open Since Year | 2007 |
| Promo2 | 1 |
| Promo2 Since Week | 13 |
| Promo2 Since Year | 2010 |
| Promo2 Jan | 1 |
| Promo2 Feb | 0 |
| Promo2 Mar | 0 |
| Promo2 Apr | 1 |
| Promo2 May | 0 |
| Promo2 Jun | 0 |
| Promo2 Jul | 1 |
| Promo2 Aug | 0 |
| Promo2 Sept | 0 |
| Promo2 Oct | 1 |
| Promo2 Nov | 0 |
| Promo2 Dec | 0 |

Below are two model explanations each from a different model where each feature value represents the importance of a feature to the model prediction, positive values indicate that as the value of this feature increases the sales values increase, and vice-versa for the negative values.

Use the explanations to answer the questions below.

**Figure 11:** Data-sample from Rossmann dataset.

Explanation A from model A

| Features | Explanation (feature importance weights) Time Step 1 |
|---|---|
| Store | -0.01 |
| Day Of Week | -0.02 |
| # Customers | 1.18 |
| Open | 0.00 |
| Promo | 0.07 |
| State Holiday | -0.01 |
| School Holiday | -0.01 |
| Year | -0.17 |
| Month | 0.02 |
| Day | 0.01 |
| Store Type | 0.01 |
| Assortment | 0.00 |
| Competition Distance | 0.00 |
| Competition Open Since Month | 0.00 |
| Competition Open Since Year | -0.01 |
| Promo2 | -0.04 |
| Promo2 Since Week | 0.00 |
| Promo2 Since Year | -0.01 |
| Promo2 Jan | 0.02 |
| Promo2 Feb | -0.10 |
| Promo2 Mar | 0.05 |
| Promo2 Apr | 0.15 |
| Promo2 May | 0.03 |
| Promo2 Jun | -0.17 |
| Promo2 Jul | -0.09 |
| Promo2 Aug | -0.01 |
| Promo2 Sept | -0.02 |
| Promo2 Oct | -0.02 |
| Promo2 Nov | -0.08 |
| Promo2 Dec | 0.05 |

Given the above input and explanation A shown above (not your own intuition) if the number of customers changes from 712 to 800 how will the model A output change?

○ Increase

○ Decrease

○ No change

**Figure 12:** Explanation from Model A and sample question. Color code is red to green representing positive value to negative value of explanation.

Given the above input and explanation B (shown below) please predict model B's output.

| Features | Explanation (feature importance weights) Time Step 1 |
|---|---|
| Store | -0.14 |
| Day Of Week | 0.47 |
| # Customers | -0.82 |
| Open | 0.13 |
| Promo | 0.10 |
| State Holiday | 0.02 |
| School Holiday | 0.05 |
| Year | 0.00 |
| Month | 0.00 |
| Day | 0.03 |
| Store Type | -0.08 |
| Assortment | -0.10 |
| Competition Distance | -0.10 |
| Competition Open Since Month | 0.56 |
| Competition Open Since Year | -0.13 |
| Promo2 | -0.13 |
| Promo2 Since Week | -0.08 |
| Promo2 Since Year | -0.01 |
| Promo2 Jan | -0.06 |
| Promo2 Feb | 0.05 |
| Promo2 Mar | 0.09 |
| Promo2 Apr | -0.28 |
| Promo2 May | 0.03 |
| Promo2 Jun | 0.00 |
| Promo2 Jul | -0.03 |
| Promo2 Aug | 0.05 |
| Promo2 Sept | 0.05 |
| Promo2 Oct | -0.12 |
| Promo2 Nov | 0.01 |
| Promo2 Dec | -0.03 |

Given the above input and explanation B shown above (not your own intuition) if the number of customers changes from 712 to 800 how will the model B output change?

○ Increase

○ Decrease

○ No change

**Figure 13:** Explanation from Model B and sample question. Color code is red to green representing positive value to negative value of explanation.

**Figure 14:** User study model trust question

