# OpenReview forum: "Interpretable Mixture of Experts"
_TMLR — Accepted by TMLR_

### Review · Reviewer_5L87 · 2023-03-21

**Summary Of Contributions:**



The authors propose an Interpretable Mixture of Experts (IME) framework that offers a variety of degrees of interpretability by using an assignment model (which could be interpretable or a black-box) to select which expert model (which could be one of a set of interpretable models or a black-box model) to use to make a prediction.  The authors evaluate IME's predictive performance on a variety of tabular and time-series datasets and run an ablation study of IME's design choices.  Further, the authors demonstrate IME's interpretability using a few examples and a user study. Overall, it appears that IME improves the Pareto frontier for interpretability and predictive performance.


**Audience:**

Yes

**Claims And Evidence:**

Yes

**Requested Changes:**



Major:
-  Many of the experiments where IME has higher predictive performance than a black-box model come from time-series datasets, where IME was given information about "past-errors."  Were the black-box models also given this information?  This seems like very useful information and, consequently, all models should have access to it in order to fairly compare their performance.
-  The User Study uses SHAP, which is a local feature attribution explanation (ie, it estimates the importance of a feature by imagining what would happen if it was “missing”), for a counterfactual task (ie, we want to know what would happen if a if a feature was “slightly changed”).  A local approximation-based explanation (eg, LIME) would have been a fairer comparison since it offers a similar type of explanation as IME with linear expert models.

Moderate:
-  When discussing simple/inherently interpretable/white-box models, there is always a question of "how hard" it is for a user to interpret the model based on the complexity of the architecture.  This is especially relevant when comparing the predictive performance of different model architectures.  As a result, it seems important to discuss the ways in which interpreting IME may be "harder" than some of the simple/inherently interpretable/white-box models that it is compared against.  For example, $\text{S-IME}_i$ requires the user to interpret an $n$-way classifier and $n$ different expert models.
-  While the synthetic datasets used for the “Identifying data distribution modes/incorrect model behavior/temporal regime changes” results make for clear exposition of these capabilities, XAI is rife with examples of methods working in very simple settings and then not working in more realistic settings.  As a result, the “real world implications” discussed at the end of these paragraphs seem under supported.
-  The User Study is presented as evaluating the quality of IME’s explanations vs the quality of post-hoc explanations.  However, the results of this experiment may be biased because I think that (please correct me if I am wrong) IME is explaining itself while SHAP is explaining a standard black-box model.  As a result, if this counterfactual task is harder for that black-box model than it is for IME (as a predictive model), the experiment would be biased towards IME.  If this understanding is correct, then this should be clarified or the claim should be revised.

Minor:
-  Please double check the bolding/italicization of the best results in the tables because there may be some errors.  For example, I think TCN has the best result for Climate 168.
-  Why isn't there an ablation for removing $L_{div}$?


**Strengths And Weaknesses:**

Strengths:
-  The IME framework provides a range of options that offer different predictive performance and interpretability characteristics.  Further, the authors do a good job of clearly and concisely describing these options.

Weaknesses (see next section for details):
-   IME’s predictive performance results for time-series datasets may be strongly influenced by having access to information about “past-errors.”
-  The claims/evidence in Section 4.4 could use some refinement/qualification to be better supported.

---

> ### Author Response · Authors · 2023-04-08
> **Response to reviewer 5L87 part 1**
>
> Thanks for all your valuable feedback that has helped us to improve our manuscript. Please see the detailed responses below and let us know if you have any further questions or comments.
>
> ### Major change requests:
>
> **Answer 1:** Past errors are indeed very useful information for the IME assignment module as they can help the assignment module choose the best expert in the next time step. For black-box models, there is no assignment operation since there is a single model (expert), so direct use of past errors wouldn’t be the same. Also, note that the past errors are only used in the assignment module; the expert itself does not take its past error as an input.
> For the requested comparison to consider using the exact set of inputs, the use of past error as the input for the downstream prediction task is possible for a single black-box model. Towards this end, we have added a new experiment on ETTm1 time-series dataset where the input to the DNN is the time-series + networks past error, the results for forecasting different horizons are shown below:
>
> | Horizon |  LSTM | Infor.| Trans. |  TCN  | LSTM+PE | Infor.+PE | Trans.+PE | TCN+PE | S-IME$_{i}$ | S-IME$_{d}$ |
> |:------------------:|:-----:|:--------:|:-----------:|:-----:|:------------------:|:----------------------:|:------------------------:|:-----------------:|:-----:|:-----:|
> | 24                 | .017 | .018    | .017       | .017 | .015              | .015                  | .016                    | .016             | .012 | .011 |
> | 48                 | .029 | .056    | .036       | .032 | .032              | .033                  | .033                    | .026             | .021 | .020 |
> | 168                | .161 | .171    | .137       | .110 | .113              | .142                  | .123                    | .168             | .047 | .044 |
>
>
> Infor. &rarr; Informer
>
> Trans. &rarr; Transformer
>
> +PE  &rarr; adding past error as an input
>
>
> We find that adding past errors helps the accuracy of DNNs but even with past errors the performance is less than that of IME. We added this experiment in the revised manuscript Appendix C.4.
>
>
> **Answer 2:** Both SHAP and LIME are post-hoc local feature importance attribution methods. When designing our user study, we chose SHAP rather than LIME, since SHAP was compared with LIME when it was originally proposed [1]. The paper showed that LIME uses a simplified input mapping that is equivalent to the approximation of the SHAP mapping (Section 4.1). They also performed a user study (Section 5) where they showed that SHAP values prove to be more consistent with human intuition than LIME. Other papers [2-5], have also shown similar results on different applications. That’s why we mainly focused on SHAP for our user study.
>
> [1] Lundberg, S.M. and Lee, S.I., 2017. A unified approach to interpreting model predictions.
>
> [2] Roberts, C.V., Elahi, E. and Chandrashekar, A., 2022. On the Bias-Variance Characteristics of LIME and SHAP in High Sparsity Movie Recommendation Explanation Tasks.
>
> [3] Misheva, B.H., Osterrieder, J., Hirsa, A., Kulkarni, O. and Lin, S.F., 2021. Explainable AI in credit risk management.
>
> [4] Liu, Y., Khandagale, S., White, C. and Neiswanger, W., 2021. Synthetic benchmarks for scientific research in explainable machine learning.
>
> [5] Moscato, V., Picariello, A. and Sperlí, G., 2021. A benchmark of machine learning approaches for credit score prediction.

---

> ### Author Response · Authors · 2023-04-08
> **Response to reviewer 5L87 part 2**
>
> ### Moderate change requests:
>
> **Answer 1:**  That’s indeed a great point. Compared to a single interpretable model, the global explanations offered by S-IME would be harder to interpret, as it would require interpretation of the assignment module as well. We have added clarification on this as suggested in the Sec. 4.4 of the revised manuscript. On the other hand, for local explanations (similar to those given by post-hoc models like LIME), the difficulty level of interpretations is the same as a single interpretable expert, as one expert for each sample is selected and the interpretability comes from analysis of the chosen expert.
>
> We also reemphasize that the interpretability-accuracy trade off is a main theme of the paper - S-IME significantly outperforms a single interpretable model, while having the extra global interpretability challenge due to the assignment module and having a similar degree of local explainability - thus it could be preferred in many situations.
>
>
> **Answer 2:** Thanks for noting this down. First, we have toned down our claims on “real world implications” as the demonstration on real world data is limited. One important thing to note is that such an analysis on real-world data would be very challenging as we wouldn’t know the ground truth for explanations or data distribution models or temporal regimes, unless we know the data generation process entirely. The complexity of the data might affect the results as the Reviewer indicates. Lastly, we note that we present the User Studies on real data, to demonstrate the quality and utility of the generated explanations.
>
>
> **Answer 3:** Yes, you are correct that IME shows its own explanations while SHAP is a post-hoc method used to explain the black-box model. IME is expected to provide better explanations since it can produce an exact explanation as the actual interpretable models behind each prediction are known and this is the main motivation of IME.
>
> In this user study we look into the “quality” from the perspective of its “utility” to the user - given an explanation, would the user be able to understand, interact and modify a model based on the explanation? If the user was given a complicated explanation even if it is the ground truth they might not be able to understand how this affects the model and in this case the explanation is useless. We show that IME offers ground-truth explanations in a simple way that can be understood by the users. We have clarified what we mean by “quality” in the revised manuscript. Another aspect we tested is how the users trust different explanations, this is a fair comparison since the users are unaware of how each explanation was generated. We found that 87% of users selected IME while only 6.5% for SHAP and 6.5 % selected neither.

---

> ### Author Response · Authors · 2023-04-08
> **Response to reviewer 5L87 part 3**
>
> ### Minor change requests:
>
>
> **Answer 1:** Thank you for catching this, we fixed the bolding of TCN in the revised manuscript and we went over all the tables and made sure they are correct.
>
> **Answer 2:** The diversity loss is more effective on time series dataset. Below is an example of MSE with and without L_div on the ETTm1 dataset - adding the diversity loss helps decrease MSE in this case. We have added these results in the Appendix of the modified manuscript section C.5.
>
> | Forecast   horizon | S-IME_i without $L_{div}$ | S-IME_i with $L_{div}$ |
> |:------------------:|:-----------------------:|:--------------------:|
> | 24                 |          0.014          |         0.012        |
> | 48                 |          0.022          |         0.021        |
> | 168                |          0.055          |         0.047        |

---

### Review · Reviewer_q2hs · 2023-03-30

**Summary Of Contributions:**

Recently XAI has received increasing attention from the community. Most current research focuses on post-hoc explanations. This article works on designing inherently interpretable models. It introduces a new interpretable modeling framework called Interpretable Mixture of Experts (IME) that is designed for tabular and time-series data and offers high accuracy comparable to DNNs while providing useful interpretability capabilities. IME consists of an assignment module and a mixture of experts, with each sample being assigned to a single expert for prediction. IME is demonstrated to be more accurate than single interpretable models and perform comparably with existing state-of-the-art DNNs in accuracy. Furthermore, IME's explanations are compared to commonly-used post-hoc explanations methods through a user study, and participants find IME's explanations more faithful and trustworthy.


**Audience:**

Yes

**Claims And Evidence:**

Yes

**Requested Changes:**

see the Strengths And Weaknesses section

**Strengths And Weaknesses:**

Strengths:
1. The proposed IME framework yields high accuracy, comparable to black-box deep neural networks (DNNs) in many cases. Besides, IME offers useful interpretability capabilities, making it suitable for high-stakes decision making.
2. IME is demonstrated to be more accurate than single interpretable models and perform comparably with existing state-of-the-art DNNs in accuracy. On most datasets, IME even outperforms black-box DNN baselines while providing faithful explanations.
3. IME's explanations are found to be more faithful and trustworthy than commonly-used post-hoc explanations methods, according to a user study.

Weaknesses:
1. The experiments section covers both classification and regression tasks; however, the notion module in Section 2.1 states that this paper focuses on the regression task.
2. Table 1 shows that MLP performs slightly better than LightGBM and XGBoost. However, in practical applications, LightGBM and XGBoost usually outperform DNN models for tabular datasets.
3. It is promising that the authors have compared their proposed intrinsically interpretable model with post-hoc explanation methods such as SHAP. The results indicate that the proposed method significantly outperforms SHAP (e.g., IME is selected in 87% of cases compared to only 6.5% for SHAP). It would be helpful if the authors could provide further explanations for the comparison results. Additionally, it would be interesting to see a quantitative comparison of the interpretability performance between the proposed method and SHAP.
4. It would be beneficial to include more explanation visualizations in the main document rather than in the appendix.

---

> ### Author Response · Authors · 2023-04-08
> **Response to reviewer q2hs**
>
> Thanks for all your valuable feedback that has helped us to improve our manuscript. Please see the detailed responses below and let us know if you have any further questions or comments.
>
> **Answer 1:** To adapt IME for classification problems, the model outputs are changed to output class probabilities instead of a single regression value. The prediction loss $L_{pred}$ is modified such that $ ||Y-F(X)||^2$ will be replaced with standard softmax cross entropy. We have added the exact formulation to the Appendix Sec. B and referred to this in the main text as well.
>
> **Answer 2:** For the Rossmann dataset we obtained the hyperparameters for CatBoost, LightGBM, and XGBoost from this benchmark: https://github.com/catboost/benchmarks/tree/master/kaggle/rossmann-store-sales. We were able to reproduce results similar to that reported in the benchmark. When using MLP we found that we can produce better results. You are correct that usually for tabular datasets, LightGBM and XGBoost outperform DNNs. However, this is not always the case - especially when the tabular datasets are large, do not suffer from distribution shifts and contain complex nonlinear patterns. Other papers also report scenarios where DNNs outperform XGBoost and LightGBM on tabular datasets, like [1] [2] [3].
>
> **Answer 3:**  Thanks for the suggestion, we have expanded our explanations. As noted, we compared post-hoc SHAP applied to a black-box model (which is MLP in this case), with IME that has intrinsic interpretability capabilities. We provided users the inputs and the models’ explanations for the given input, and asked them, given the model’s explanation how they think the model would behave when the input is changed. When provided with IME explanations, we found that participants can predict model behavior with an accuracy of 69% vs. 42% when MLP+SHAP was used. We then asked the users which model do you trust more. IME was selected by 87% of users compared to only 6.5% for SHAP as you mentioned. We hope these clarify the comparison results, and we would be happy to provide more details. Finally, we note that this user study is conducted on a real-world retail dataset, for which the ground truth explanations do not exist. Therefore our quantitative comparisons are based on statistics obtained from user inputs.
>
> **Answer 4:** As requested we have moved sample-wise expert interpretability figures to the main text.
>
> [1] Rishabh Agarwal, Nicholas Frosst, Xuezhou Zhang, Rich Caruana, and Geoffrey E Hinton. “Neural additive models: Interpretable machine learning with neural nets”
>
> [2] Sercan Arik and Tomas Pfister. “TabNet: Attentive Interpretable Tabular Learning”
>
> [3] Yury Gorishniy, Ivan Rubachev, Valentin Khrulkov, Artem Babenko. “Revisiting Deep Learning Models for Tabular Data”

---

### Review · Reviewer_mS9v · 2023-03-31

**Summary Of Contributions:**

This paper proposes Interpretable Mixture of Experts (IME) for modeling tabular and time-seres data that combines multiple interpretable models (e.g., linear regression) in a mixture of experts. IME preserves accuracy, and provides useful interpretability capabilities. Moreover, the authors present multiple options for IME, including single-level assignment and hierarchical assignnment options. Through experiments, the paper shows that IME outperforms other interpretable architectures, and offers easier-to-interpret explanations.

**Audience:**

No

**Broader Impact Concerns:**

The proposed method appears to be applicable only in specific domains where linear regression performs well, such as time series or tabular data. Moreover, rather than introducing a new interpretation methodology, the paper employs a mixture of experts, which may not have a significant impact, since interpretable models are used as experts or assignment modules.

**Claims And Evidence:**

Yes

**Requested Changes:**

Overall, a major drawback is the limited novelty of the proposed method. The results indicating that experts specialized in different data are not novel, and do not appear to be impactful. I believe that it would be preferable to introduce a methodology that can be interpreted, irrespective of the structure of experts and assignment modules, in order to support the claim of having an interpretable mixture of experts.

**Strengths And Weaknesses:**

The paper has several strengths, one of which is its ability to preserve accuracy while providing interpretability. The proposed framework, Interpretable Mixture of Experts (IME), achieves this by combining simple and interpretable models, such as linear regression. The paper also presents multiple options for IME to address interpretability needs of particular applications, including single-level and hierarchical assignment options.

However, the paper's main weakness is its limited approach to enhancing interpretability. While the authors claim to introduce a new class of interpretable models, the use of linear regression for experts or assignment modules is not entirely novel, as mixture of experts have already been widely used in various fields. Furthermore, while the authors believe that using linear regression is preferable because of its interpretability, I am uncertain about why this would be superior to using deep neural networks (DNNs). The usefulness of knowing the trained parameters of linear regression does not seem very useful, and given the restricted capacity of linear regression, it seems that the proposed method may have limited applicability. Moreover, the paper fails to sufficiently demonstrate the benefits of enhancing interpretability in their experiments. Overall, while the paper presents the interpretable mixture of experts, further exploration and innovation in enhancing interpretability are needed to fully realize its potential.

---

> ### Author Response · Authors · 2023-04-08
> **Response to reviewer mS9v part 1**
>
> Thanks for all your valuable feedback that has helped us to improve our manuscript. Please see the detailed responses below and let us know if you have any further questions or comments.
>
> >**Reviewer's comment:** However, the paper's main weakness is its limited approach to enhancing interpretability. While the authors claim to introduce a new class of interpretable models, the use of linear regression for experts or assignment modules is not entirely novel, as mixture of experts have already been widely used in various fields.
>
>
> **Answer:**  We agree that the mixture of experts have been around since the '90s [1], and have been widely used since then. For example, Shazeer et al. [2] used a mixture of experts in DNNs by stacking a mixture of experts layer between LSTM layers. Shazeer et al.[3], Lepikhin et al.[4] and Fedus et al.[5] incorporated a mixture of experts layer into transformers. Pradier et al. [6] uses a mixture of experts in a human-ML setting where the assignment module in the mixture of experts depends on human-based rules, and the experts themselves are black-box DNNs. However, the motivations of them and the problem scenario differ from our work as they did not consider a mixture of experts framework to improve interpretability and for tabular and time-series prediction problems. Although the idea of a mixture of experts itself is not novel at all, the way we consider it for interpretable tabular and time-series prediction is novel, as it has not been done before, and it indeed requires careful design of the architecture and training objectives. We present Ablation Studies to compare IME to other conventional mixtures of experts variants (in Table 14), and show that IME significantly outperforms them, with its carefully designed components. We have revised our claims in the Introduction about novelty to be more specific about our contribution.
>
>
> [1] Robert A Jacobs, Michael I Jordan, Steven J Nowlan, and Geoffrey E Hinton. Adaptive mixtures of local experts. Neural computation, 1991.
>
> [2] Noam Shazeer, Azalia Mirhoseini, Krzysztof Maziarz, Andy Davis, Quoc Le, Geoffrey Hinton, and Jeff Dean. Outrageously large neural networks: The sparsely-gated mixture-of-experts layer.
>
> [3] Noam Shazeer, Youlong Cheng, Niki Parmar, Dustin Tran, Ashish Vaswani, Penporn Koanantakool, Peter Hawkins, HyoukJoong Lee, Mingsheng Hong, Cliff Young, et al. Mesh-tensorflow: Deep learning for supercomputers.
>
> [4] Dmitry Lepikhin, HyoukJoong Lee, Yuanzhong Xu, Dehao Chen, Orhan Firat, Yanping Huang, Maxim Krikun, Noam Shazeer, and Zhifeng Chen. Gshard: Scaling giant models with conditional computation and automatic sharding.
>
> [5] William Fedus, Barret Zoph, and Noam Shazeer. Switch transformers: Scaling to trillion parameter models with simple and efficient sparsity.
>
> [6] Rishabh Agarwal, Nicholas Frosst, Xuezhou Zhang, Rich Caruana, and Geoffrey E Hinton. Neural additive models: Interpretable machine learning with neural nets.
>
> [6] Melanie F Pradier, Javier Zazo, Sonali Parbhoo, Roy H Perlis, Maurizio Zazzi, and Finale Doshi-Velez. Preferential mixture-of-experts: Interpretable models that rely on human expertise as much as possible.

---

> ### Author Response · Authors · 2023-04-08
> **Response to reviewer mS9v part 2**
>
>
> >**Reviewer's comment:** Furthermore, while the authors believe that using linear regression is preferable because of its interpretability, I am uncertain about why this would be superior to using deep neural networks (DNNs). The usefulness of knowing the trained parameters of linear regression does not seem very useful, and given the restricted capacity of linear regression, it seems that the proposed method may have limited applicability.
>
> **Answer:**  First and foremost, we would like to clarify that experts in IME are not limited to linear regression models, but they can be any interpretable model. IME is a canonical framework and is model agnostic. We also present results with soft decision trees for example, constituting another interpretable model architecture that can be used within IME. We have clarified the point that IME is agnostic to the type of the expert used.
>
> DNNs might obtain high accuracy yet they lack interpretability - as their decision making is based on complex nonlinear interactions between the input features. To shed light into the decision making of DNNs, post-hoc explainability methods are needed, but those do not come with guarantees on the correctness of explanations and often such methods are unreliable [1-5], and that’s why in most high-stakes decision making scenarios like in Healthcare or Finance avoid DNNs.
>
> Interpretable models (like linear regression) on the other hand, provide explanations that are guaranteed to represent decision making rationale in some form (like how the input features are combined and mapped to the output), by design. In that way, they differ from DNNs, and especially in high-stakes decision making scenarios, they are commonly used. As the Reviewer also mentioned, interpretable models have limited capacity due to their design, and they yield much lower accuracy in most scenarios.
>
> The motivation of IME is to bring the best of both worlds - while keeping the most useful aspects of globally interpretable models (e.g. still being able to precisely describe how the input features are mapped to the output when linear regression assigners and experts are used), IME can push the accuracy of simple interpretable models significantly. In other words, in many scenarios, IME enables obtaining interpretable models that achieve almost similar accuracy with DNNs (please see Tables 1-3), with their explanations coming from the design (not via post-hoc approximations) and being more robust.
>
> We hope these clarify our motivations and contributions, as these are some key aspects behind our work.
>
> [1] Julius Adebayo, Justin Gilmer, Michael Muelly, Ian Goodfellow, Moritz Hardt, and Been Kim. Sanity checks for saliency maps.
>
> [2] Sara Hooker, Dumitru Erhan, Pieter-Jan Kindermans, and Been Kim. A benchmark for interoperability methods in deep neural networks.
>
> [3] Amirata Ghorbani, Abubakar Abid, and James Zou. Interpretation of neural networks is fragile.
>
> [4] Aya Abdelsalam Ismail, Mohamed Gunady, Héctor Corrada Bravo, and Soheil Feizi. Benchmarking deep learning interpretability in time series predictions.
>
> [5] Srinivas, S. and Fleuret, F., 2020. Rethinking the role of gradient-based attribution methods for model interpretability.

---

> ### Author Response · Authors · 2023-04-08
> **Response to reviewer mS9v part 3**
>
>
> >**Reviewer's comment:** Moreover, the paper fails to sufficiently demonstrate the benefits of enhancing interpretability in their experiments. Overall, while the paper presents the interpretable mixture of experts, further exploration and innovation in enhancing interpretability are needed to fully realize its potential.
>
> **Answer:** Highly accurate models are desired for tasks on complex tabular and time-series data, for which recently DNNs have shown state-of-the-art performance. However, DNNs are black-box models that the complex nonlinear decision making process can not be interpreted by humans. To address this over a dozen explainability methods [1-17] have been proposed to understand how DNNs produce a prediction given a specific input. Unfortunately, such methods can only offer a local approximation of how DNNs work and have proven to be unreliable [18-22]. In addition, for a given application, it is not clear which method should be selected [23,24] since different methods produce different explanations. There are numerous high-stakes decision making applications where black-box models are not acceptable because users would like to build trust in a model’s decision by understanding it completely, or the model builders would like to diagnose issues quickly to improve the models, or certain regulatory standards need to be satisfied to avoid fairness or legality issues. All in all, there is a significant need for interpretable modeling, and simple black-box models cannot penetrate into such use cases, and simple interpretable models (like linear regression or decision trees) are still being used. Correspondingly, we are motivated by the fundamental questions - are there family of interpretable models that can be better than conventional interpretable models; and can we enable any knob that can trade off between interpretability and accuracy; and can there be any intermediate forms of interpretability like local, that can still be useful for the aforementioned purposes without degrading the accuracy? We hope these clarify the motivations on why enhancing interpretability is beneficial.
>
> Specifically for IME in our paper, we have presented the quality and utility of explanations in numerous ways in Sec. 4.4. For example, obtaining global interpretability as in the form of Algorithm 2 can simply unlock the use of ML in highly-regulation sensitive applications. Identifying data distribution modes and temporal regime changes are useful capabilities that the model builders can benefit from. Lastly and most importantly, we indeed have a user study where we specifically ask the questions around usefulness and quality of the explanations, and we show very strong results on IME explanations.

---

> > ### Author Response · Authors · 2023-04-08
> > **citations**
> >
> > [1] M. Ribeiro, S. Singh, and C. Guestrin. Anchors: High-precision model-agnostic explanations.
> >
> > [2] K. Simonyan, A. Vedaldi, and A. Zisserman. Deep inside convolutional networks: Visualising image classification models and saliency maps.
> >
> > [3] David Baehrens, Timon Schroeter, Stefan Harmeling, Motoaki Kawanabe, Katja Hansen, and Klaus-Robert M˜Aˇzller. How to explain individual classification decisions.
> >
> > [4] Jost Tobias Springenberg, Alexey Dosovitskiy, Thomas Brox, and Martin Riedmiller. Striving for simplicity: The all convolutional net.
> >
> > [5] Pieter-Jan Kindermans, Kristof Sch¨utt, Klaus-Robert M¨uller, and Sven D¨ahne. Investigating the influence of noise and distractors on the interpretation of neural networks.
> >
> > [6] Mukund Sundararajan, Ankur Taly, and Qiqi Yan. Axiomatic attribution for deep networks.
> >
> > [7] Daniel Smilkov, Nikhil Thorat, Been Kim, Fernanda Vi´egas, and Martin Wattenberg. Smoothgrad: removing noise by adding noise.
> >
> > [8] Ramprasaath R Selvaraju, Michael Cogswell, Abhishek Das, Ramakrishna Vedantam, Devi Parikh, and Dhruv Batra. Grad-cam: Visual explanations from deep networks via gradient-based localization.
> >
> > [9] Sebastian Bach, Alexander Binder, Gr´egoire Montavon, Frederick Klauschen, Klaus-Robert M¨uller, and Wojciech Samek. On pixel-wise explanations for non-linear classifier decisions by layer-wise relevance propagation.
> >
> > [10] Avanti Shrikumar, Peyton Greenside, and Anshul Kundaje. Learning important features through propagating activation differences.
> >
> > [11] Scott M Lundberg and Su-In Lee. A unified approach to interpreting model predictions.
> >
> > [12] Matthew D Zeiler and Rob Fergus. Visualizing and understanding convolutional networks.
> >
> > [13] Harini Suresh, Nathan Hunt, Alistair Johnson, Leo Anthony Celi, Peter Szolovits, and Marzyeh Ghassemi. Clinical intervention prediction and understanding using deep networks.
> >
> > [14] Marco Tulio Ribeiro, Sameer Singh, and Carlos Guestrin. "Why should I trust you?" Explaining the predictions of any classifier.
> >
> > [15] Javier Castro, Daniel G´omez, and Juan Tejada. Polynomial calculation of the shapley value based on sampling.
> >
> > [16] Vitali Petsiuk, Abir Das, and Kate Saenko. Rise: Randomized input sampling for explanation of black-box models.
> >
> > [17] Gr´egoire Montavon, Sebastian Lapuschkin, Alexander Binder, Wojciech Samek, and Klaus-Robert M¨uller. Explaining nonlinear classification decisions with deep taylor decomposition.
> >
> > [18] Julius Adebayo, Justin Gilmer, Michael Muelly, Ian Goodfellow, Moritz Hardt, and Been Kim. Sanity checks for saliency maps.
> >
> > [19] Sara Hooker, Dumitru Erhan, Pieter-Jan Kindermans, and Been Kim. A benchmark for interoperability methods in deep neural networks.
> >
> > [20] Amirata Ghorbani, Abubakar Abid, and James Zou. Interpretation of neural networks is fragile.
> >
> > [21] Aya Abdelsalam Ismail, Mohamed Gunady, Héctor Corrada Bravo, and Soheil Feizi. Benchmarking deep learning interpretability in time series predictions.
> >
> > [22] Srinivas, S. and Fleuret, F., 2020. Rethinking the role of gradient-based attribution methods for model interpretability.
> >
> > [23] Krishna, S., Han, T., Gu, A., Pombra, J., Jabbari, S., Wu, S. and Lakkaraju, H., 2022. The disagreement problem in explainable machine learning: A practitioner's perspective.
> >
> > [24] Neely, M., Schouten, S.F., Bleeker, M.J. and Lucic, A., 2021. Order in the court: Explainable ai methods prone to disagreement.

---

> ### Author Response · Authors · 2023-04-08
> **Response to reviewer mS9v part 4**
>
> >**Reviewer's comment** Overall, a major drawback is the limited novelty of the proposed method. The results indicating that experts specialized in different data are not novel, and do not appear to be impactful. I believe that it would be preferable to introduce a methodology that can be interpreted, irrespective of the structure of experts and assignment modules, in order to support the claim of having an interpretable mixture of experts.
>
> **Answer:** We appreciate the feedback on novelty, and we have revised our Introduction section to better clarify our novel contributions. If the Reviewer thinks we have a specific overclaim on novelty, we would be happy to tone down our claims.
>
> To recap the main points again, we believe IME is novel as it is the first proposal of a mixture of experts framework for interpretability purposes for tabular and time-series prediction problems (that constitute the two most common problems of the addressable AI market [1]). We clearly show (Table 4 and 14) that straightforward extension of conventional mixture of experts' ideas wouldn’t be effective in achieving high accuracy and careful design in the architecture and loss function are needed. Many proposed components of IME are judiciously chosen and methodological innovations play a key role in the competitive results, as our Ablation Studies show. For example, the use of past errors to improve assignment accuracy has not been proposed before, or the way we combine multiple loss functions does not exist in other papers. The specificity of the experts helps produce high-accuracy predictions and since simple interpretable models have low capacity they will not be able to model large data while maintaining accuracy. The overall idea revolves around having a mixture of interpretable experts where each expert can be easily understood by humans, for a given data sample as the prediction comes from a single expert. Since we provide interpretable local explanations for each sample, our method is in fact an interpretable mixture of experts. We demonstrate the quality and utility of the produced explanations in Sec. 4.4 both on synthetic and real-world data, including user studies.
>
> [1] Chui, M., Manyika, J., Miremadi, M., Henke, N., Chung, R., Nel, P. and Malhotra, S., 2018. Notes from the AI frontier: Insights from hundreds of use cases. McKinsey Global Institute, 2.

---

### Author Response · Authors · 2023-04-08
**Submitted the revised manuscript and detailed responses to each reviewer**

Thanks for all your valuable feedback that has helped us improve our manuscript. We uploaded a modified manuscript to address the changes requested by the reviewers, we highlighted changes in the modified manuscript by coloring them in red.  Please see the modified manuscript and detailed responses below and let us know if you have any further questions or comments.

---

### Decision · Action_Editors · 2023-05-10

**Recommendation:** Accept as is

**Comment:**

The authors propose a framework for interpretable machine learning based on mixtures of experts. This enables them, for settings like time-series and tabular data, to maintain good accuracy---comparable to non-interpretable models---while also providing good explanations.

All reviewers found a great deal to like, and I felt the same way. The proposed technique is clearly of interest and could be useful in a variety of settings. The authors provide a solid amount of evidence supporting their claims, including a user study. The authors have also improved the writing and positioning as a result of feedback from reviewers, resulting in a clear and well-written work.

The only drawbacks mentioned by the reviewers are limited novelty and potentially limited impact. The first of these is not a focus, so I will not attempt to evaluate it (my sense is that there are multiple innovations so even this bar would be cleared easily). The second concern is not a problem, as the reviewer notes that the proposed approach does impact important areas like time-series and tabular data.

Altogether this is solid work.

**Audience:**

Yes, interpretable machine learning is of interest to a large part of the TMLR audience.

**Claims And Evidence:**

Yes, the authors have solid evidence for the claims made.